# Sex differences in learning from exploration

Cathy S Chen[1], Evan Knep[1], Autumn Han[1], R Becket Ebitz[2]*, Nicola M Grissom[1]*

[1]Department of Psychology, University of Minnesota, Minneapolis, United States; [2]Department of Neurosciences, University of Montreal, Quebec, Canada

**Abstract** Sex-based modulation of cognitive processes could set the stage for individual differences in vulnerability to neuropsychiatric disorders. While value-based decision making processes in particular have been proposed to be influenced by sex differences, the overall correct performance in decision making tasks often show variable or minimal differences across sexes. Computational tools allow us to uncover latent variables that define different decision making approaches, even in animals with similar correct performance. Here, we quantify sex differences in mice in the latent variables underlying behavior in a classic value-based decision making task: a restless two-armed bandit. While male and female mice had similar accuracy, they achieved this performance via different patterns of exploration. Male mice tended to make more exploratory choices overall, largely because they appeared to get 'stuck' in exploration once they had started. Female mice tended to explore less but learned more quickly during exploration. Together, these results suggest that sex exerts stronger influences on decision making during periods of learning and exploration than during stable choices. Exploration during decision making is altered in people diagnosed with addictions, depression, and neurodevelopmental disabilities, pinpointing the neural mechanisms of exploration as a highly translational avenue for conferring sex-modulated vulnerability to neuropsychiatric diagnoses.

**\*For correspondence:**
rebitz@gmail.com (RBecketE);
ngrissom@umn.edu (NMG)

**Competing interest:** The authors declare that no competing interests exist.

## Editor's evaluation

Following inclusion of new modeling and data presentation, authors have more clearly demonstrated that equivalent performance is seen across males and females in terms of reward rate, yet achieved via different successful strategies. This is an important contribution to the growing literature on sex differences in reinforcement learning.

## Introduction

Almost every neuropsychiatric condition shows sex and/or gender biases in risk, presentation, etiology, and prognosis (*Green et al., 2019*; *Grissom and Reyes, 2019*; *Shansky, 2019*). This raises the possibility that sex-modulated biological mechanisms could modulate cognitive processes that confer vulnerability and/or resilience to mental health challenges. However, sex differences in cognitive task performance can be difficult to detect and even more variable than would be expected given the non-dichotomous, overlapping impacts of sex mechanisms on cognition (*Maney, 2016*). An underrecognized source of variability in cognitive tasks is that there can be multiple ways to achieve the same level of performance on the primary dependent variables used to assess these tasks, such as 'number of correct responses'. This means that equivalent levels of performance could mask individual differences in how males and females are solving the same problem. Indeed, we have recently shown that examining the latent strategies underlying task performance --rather than differences in final performance--can reveal that individual males and females can take very different strategic paths

**eLife digest** When faced with a decision to make, humans and other animals reflect on past experiences of similar situations to choose the best option. However, in an uncertain situation, this decision process requires balancing two competing priorities: exploiting options that are expected to be rewarding (exploitation), and exploring alternatives that could be more valuable (exploration).

Decision making and exploration are disrupted in many mental disorders, some of which can differ in either presentation or risk of development across women and men. This raises the question of whether sex differences in exploration and exploitation could contribute to the vulnerability to these conditions. To shed light on this question, Chen et al. studied exploration in male and female mice as they played a video game.

The mice had the option to touch one of two locations on a screen for a chance to win a small reward. The likelihood of success was different between the two options, and so the mice were incentivized to determine which was the more rewarding button. While the mice were similarly successful in finding rewards regardless of sex, on average male mice were more likely to keep exploring between the options while female mice more quickly gained confidence in an option. These differences were stronger during uncertain periods of learning and exploration than when making choices in a well-known situation, indicating that periods of uncertainty are when the influence of sex on cognition are most visible.

However, not every female or male mouse was the same – there was as much variability within a sex as was seen between sexes. These results indicate that sex mechanisms, along with many other influences cause individual differences in the cognitive processes important for decision making. The approach used by Chen et al. could help to study individual differences in cognition in other species, and shed light on how individual differences in decision-making processes could contribute to risk and resilience to mental disorders.

to the learning of action-outcome associations (*Chen et al., 2021b*). Here, we applied computational tools to characterize sex differences in the latent variables underlying behavior to understand sex differences in a key cognitive process regulating reward-guided behaviors: balancing exploration and exploitation.

In an uncertain world, we must balance two goals: exploiting rewarding options when they are available, but also exploring alternatives that could be more rewarding or provide new information about the world. Too little exploration makes behavior inflexible and perseverative. Too much makes it impossible to sustain rewarding behaviors. Exploration is dysregulated in numerous neuropsychiatric disorders (*Addicott et al., 2017*; *Wilson et al., 2021*), many of which are also sex-biased (*Green et al., 2019*; *Grissom and Reyes, 2019*; *Shansky, 2019*). This suggests that sex differences in exploration and exploitation could contribute to sex-linked vulnerability to these conditions, though we do not yet understand how exploration and exploitation differ with sex. Because exploration is a major source of errors in task performance more broadly (*Ebitz et al., 2019*; *Pisupati et al., 2019*), sex-differences in exploration could contribute to performance differences and variability seen across tasks and speciess (*Grissom and Reyes, 2019*; *van den Bos et al., 2013*).

To examine whether there are sex differences in exploration, we trained male and female mice on a classic explore/exploit task, a spatial restless two-armed bandit (*Ebitz et al., 2018*). Males showed higher levels of exploration than females throughout the task. This was because males were more likely to get 'stuck' in extended periods of exploration before committing to a favored choice. On the other hand, females showed elevated reward learning specifically during bouts of exploration, making exploratory trials more informative, which allowed them to start exploiting a favored choice earlier than males. Together, these results demonstrate that while the overall performance was similar, males and females exhibited different patterns of exploration while interacting with the same uncertain environment.

# Results

Age-matched male and female wild-type mice (n = 32, 16 per sex, strain B6129SF1/J, powered to detect differences in decision making *Chen et al., 2021b*) were trained to perform a restless two-armed spatial bandit task in touch-screen operant chambers (*Figure 1A*). Animals were presented with two physically identical targets (squares) on the left and right of the screen each trial and indicated their choices by nose poking at one of two target locations. Each location offered some probability of reward, which changed slowly and randomly across trials, and independently across targets. The dynamic reward contingencies encouraged the animals to constantly balance exploration and exploitation. The animals had to exploit a good option when it was found, but also occasionally explore the other option, whose drifting values meant that it could become better at any time. Mice performed two repetitions of four consecutive sessions of the restless bandit task, measuring eight sessions in total. Each session consisted of 300 trials.

It is worth noting that unlike other versions of bandit tasks such as the reversal learning task, in the restless bandit task, animals were encouraged to learn about the most rewarding choice(s). There is no asymptotic performance during the task because the reward probability of each choice constantly changes. The performance is best measured by the amount of obtained reward. Prior to data collection, both male and female mice had learned to perform this task in the touchscreen operant chamber. To examine whether mice had learned the task, we first calculated the average probability of reward acquisition across sessions in males and females (*Figure 1—figure supplement 1A*). There was no significant change in the reward acquisition performance across sessions in both sexes, demonstrating that both males and females have learned to perform the task and had reached an asymptotic level of performance (two-way repeated measure ANOVA, main effect of session, p = 0.71). Then we examine two other primary behavioral metrics across sessions that are associated with learning: response time and reward retrieval time (*Figure 1—figure supplement 1A,C*). Response time was calculated as the time elapsed between the display onset and the time when the nose poke response was completed. Reward retrieval time was measured as the time elapsed between nose-poke response and magazine entry for reward collection. There was no significant change in response time (two-way repeated measure ANOVA, main effect of session, p = 0.39) and reward retrieval time (main effect of session, p = 0.71) across sessions in both sexes, which again demonstrated that both sexes have learned how to perform the task. Since both sexes have learned to perform the task prior to data collection, variabilities in task performance are results of how animals learned and adapted their choices in response to the changing reward contingencies.

To examine task performance, we first calculated the average probability of reward obtained in males and females. Because reward schedules were stochastic, sessions could differ slightly in the amount of reward that was available. We therefore compared performance against the average probability of reward if chosen randomly within each session. Regardless of sex, mice were able to earn reward more frequently than chance (*Figure 1B*, two-way ANOVA, F(1, 60) = 228.9, p < 0.0001). There was no significant sex difference in the probability of rewards acquired above chance (*Figure 1C*, main effect of sex, F(1, 30) = 0.05, p = 0.83). While the mean of percent reward obtained did not differ across sexes, we consider the possibility that the distribution of reward acquisition in males and females might be different. We conducted the Kolmogorov-Smirnov (KS) test, which takes into account not only the means of the distributions but also the shapes of the distributions. The KS test suggested that males and females are not just not significantly different in their reward acquisition performance (Kolmogorov-Smirnov D = 0.1875, p = 0.94), but that males and females have the same distributions for reward acquisition. This result demonstrates equivalently strong understanding and identical performance of the task in males and females.

Similar levels of accuracy do not require or imply a similar approach to the task. Our previous study suggested that males and females could achieve similar learning performance via divergent decision making strategies (*Chen et al., 2021b*). However, different strategies might take different amounts of time to execute (*Chen et al., 2021b*; *Filipowicz et al., 2019*; *Kool et al., 2010*; *Kurdi et al., 2019*). Therefore, we examined the response time, which was calculated as time elapsed between choice display onset and nose poke response as recorded by the touchscreen chamber, in both males and females. If males and females had adopted different strategies here, then we might expect response time to systematically differ between males and females, despite the similarities in learning performance. Indeed, females responded significantly faster than did males (*Figure 1C*, main effect of sex, t

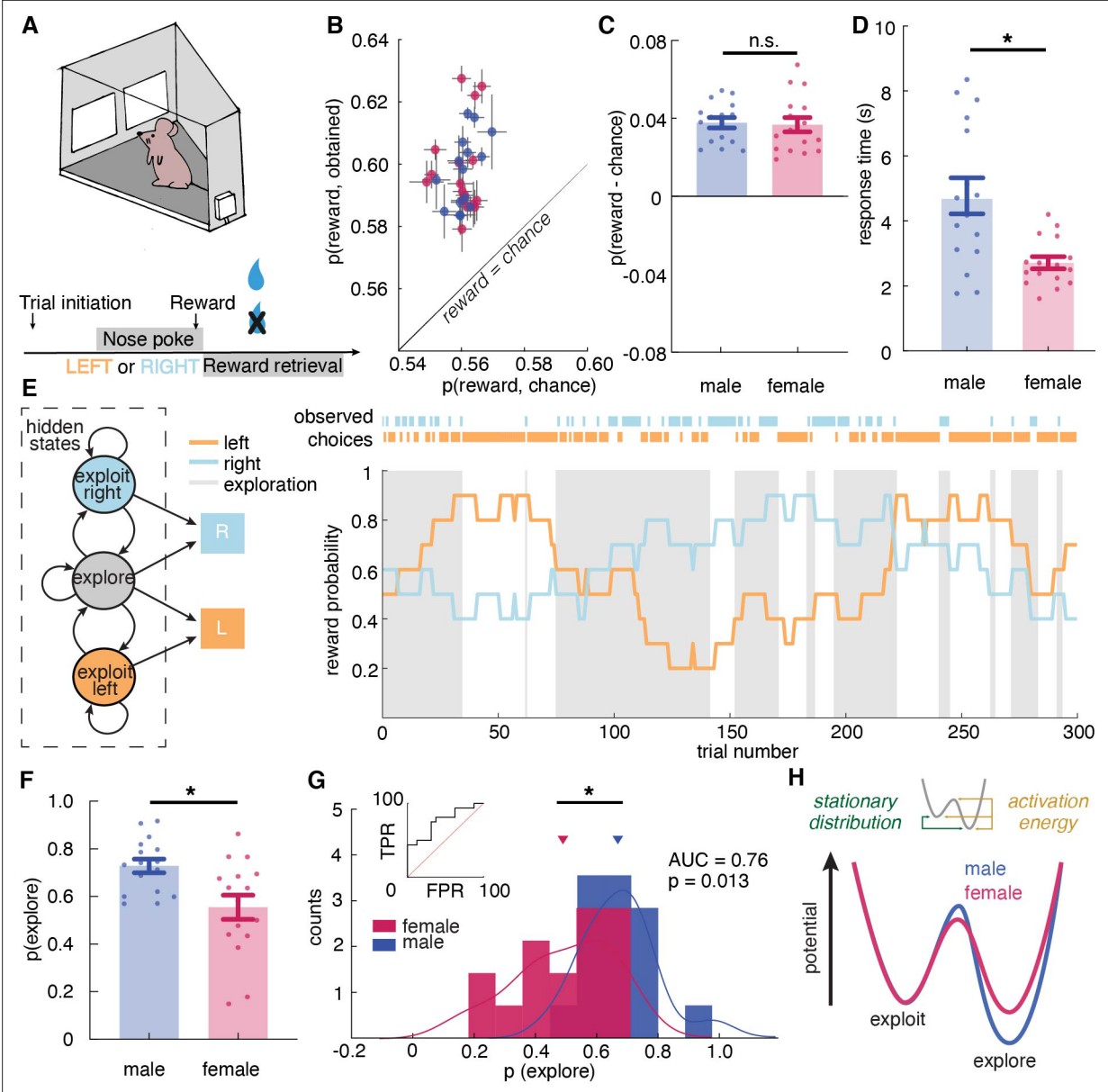

**Figure 1.** Male and female mice showed different exploratory strategies in a restless bandit task - males explored more than females, and they explored for longer periods of time once started. (**A**) Schematic of the mouse touchscreen chamber with the restless two-armed bandit task and trial structure. (**B**) Average probability of obtaining reward compared to the chance probability of reward across individuals (dots). (**C**) Average probability of obtaining reward compared to the chance probability of reward across sexes. (**D**) Average response time across sexes. Females responded significantly faster than did males. (**E**) (left) A hidden Markov model that labeled exploration and exploitation as latent goal states underlying observed choices. This model includes an exploitation state for each arm and an exploration state where the subject chooses one of the arms randomly. (right) Reward probabilities (lines) and choices (dots) for 300 example trials for a given mouse. Shaded areas highlight explore-labeled choices. (**F, G**) Average (**F**) and distribution (**G**) of the percentage of Hidden Markov Model (HMM)-labeled exploratory trials in females and males. (**H**) Dynamic landscape of the fitted HMMs for males and females. The model fit to males had deeper exploratory states, with higher activation energy between the states. * indicates p < 0.05. Graphs depict mean ± SEM across animals.

The online version of this article includes the following figure supplement(s) for figure 1:

**Figure supplement 1.** Male and female mice had reached asymptotic performance.

**Figure supplement 2.** Two time constants combined best describe the rate of switching choices in animals' choice behavior and Hidden Markov model validation.

(30) = 3.52, p = 0.0014), suggesting that decision making computations may differ across sexes and, if so, that the strategies that tended to be used by females resulted in faster choice response time than those used by males.

## A hidden Markov Model (HMM) identifies distinct features of exploratory and exploitative choices in mice

Despite similar performance, response time differences suggested that males and females employed different strategies in this task. One possible difference was sex differences in the level of exploration. Prior research has shown that exploratory choices take longer than exploitative choices (*Ebitz et al., 2019*; *Ebitz et al., 2018*). Therefore, perhaps males took longer to make a choice because a greater proportion of their choices were exploratory. To test this hypothesis, we first need a method to label each choice as an exploratory choice or an exploitative choice. In some previous studies, reinforcement learning (RL) models were used to quantify exploration (*Cinotti et al., 2019*; *Daw et al., 2006*; *Ishii et al., 2002*; *Pearson et al., 2009*) via labeling choices that deviate from model values as exploratory. This approach is based on the rationale that exploration is a non-reward maximizing goal. However, a non-reward maximizing goal would produce choices that are orthogonal to reward value, not errors of reward maximization (*Averbeck et al., 2017*; *Ebitz et al., 2018*). Therefore, recent studies have turned to an approach, which models exploration as a latent state underlying behavior via a Hidden Markov model (HMM), rather than inferring it from assumptions about values and learning (*Ebitz et al., 2019*; *Ebitz et al., 2018*; *Muller et al., 2019*).

The HMM method has not previously been used to quantify exploration in mice, so we first asked whether it was appropriate here. The method works because sequences of exploratory decisions look very different from exploitative ones, at least in reinforcement learning agents and primates (*Ebitz et al., 2018*). When agents exploit, they repeatedly sample the same option, switching only very rarely. However, because exploration requires investigatory samples, runs of exploratory choices tend not to repeat the same option. They tend to switch far more frequently, closer to what we would expect from random samples from the environment (*Ebitz et al., 2018*). Therefore, if mice were alternating between exploration and exploitation in this task, we would expect to see evidence of two distinct patterns of switching in their behavior. Indeed, choice run durations (i.e. the distribution of inter-switch intervals) were parsimoniously described as a mixture of two different patterns: one regime where choices switched quickly (mean switching time = 1.7 trials, compared to random choices at two trials; 80 % of choice runs) and one regime where they changed slowly (mean switching time = 6.8 trials; *Figure 1—figure supplement 2A*). Thus, mice had evidence of fast-switching (putatively explore) and slow-switching (putatively exploit) regimes in their behavior. Note that explore-labeled choices are more likely to also be switch choices, but not all explore-labeled choices are switches, and not all exploit-labeled choices are stay decisions.

To determine whether the novel HMM method produced more accurate labels than the previous RL method, we conducted a side-by-side comparison to examine how well each set of labels accounted for behavior. We first examined the correlation between explore-exploit states inferred by the HMM model and the RL model. We calculated the tetrachoric correlation between HMM-inferred and RL-inferred states (*Figure 1—figure supplement 2B*). The tetrachoric correlation ($r_{tet}$) is specifically used to measure rater agreement for binary data and reveals how strong the association is between labels by two methods. The mean correlation was 0.42 with a standard deviation of 0.14, which is medium level agreement.

Next, to examine whether the states inferred by these models also produced differences in behavioral metrics other than choices, we computed average response time for explore trials and exploit trials labeled by the RL model and HMM model. The result suggested that response time was significantly longer during HMM-inferred exploration than exploitation (paired t-test, t(31) = 3.66, p = 0.0009), which is consistent with previous findings that exploration slows down decision making (*Ebitz et al., 2018*). Like HMM-inferred states, RL inferred explore-exploit states showed similar effects on response time - response time was significantly longer during exploration than exploitation (paired t-test, t(31) = 2.08, p = 0.046). However, the effect size of HMM labels on response time was over twice as big as that of RL labels (HMM: $R^2$ = 0.30; RL: $R^2$ = 0.12).

Finally, we calculated the standardized regression coefficients to measure how much of the response time is explained by states labeled by HMM model and RL model (*Figure 1—figure supplement 2C*).

The result suggested that the HMM-inferred states explained significantly more variance in response time than the RL-inferred states in explaining response time. The HMM allows us to make statistical inferences about the probability that each choice was due to exploration or exploitation via modeling these as the latent goal states underlying choice (see Materials and methods). Because this approach to infer exploration is agnostic to the generative computations and depends only on the temporal statistics of choices (*Ebitz et al., 2020*; *Ebitz et al., 2019*; *Ebitz et al., 2018*; *Wilson et al., 2021*), it is particularly ideal for circumstances like this one, where we suspect that the generative computations may differ across groups.

Since various factors could influence state of the next trial, we considered a simple two parameter HMM that models only two states (exploration and exploitation), a four-parameter input-output HMM (ioHMM) that allows reward outcome to influence the probability of transitioning between states, and a four-parameter unrestricted HMM with no promoter tying (ntHMM) that allows biased exploitation (see Materials and methods). The model comparisons have shown that the two parameter HMM was the simplest, most interpretable, and best fit model (AIC: 2 parameter HMM, AIC = 2976.1; ioHMM, AIC = 3117; ntHMM, AIC = 3101.5, see more statistics reported in Materials and methods). Therefore, we selected the simple two-parameter HMM to infer the likelihood that each choice was part of the exploratory regime, or the exploitative one (see Materials and methods). To evaluate the face validity of the HMM labels, we asked whether HMM-labeled exploratory choices matched the normative definition of the term. First, by definition, exploration is a pattern of non-reward-maximizing choices whose purpose is learning about rewards. This means exploratory choices should be (1) orthogonal to reward value, and (2) exhibit enhanced reward learning.

Explore-labeled choices were non-reward-maximizing: they were orthogonal to reward value (*Figure 1—figure supplement 1D*; the average value of choices chosen during exploration was not different from chance; one sample t-test, $t(10) = 0.16$, $p = 0.87$). Reward learning was also elevated during exploration. During HMM-labeled exploratory states, the outcome of choices had more influence on the subsequent decision - animals were more likely to stay with the same choice if rewarded and switch if not rewarded (*Figure 1—figure supplement 1F*, two-way RM ANOVA, interaction term, $F(1,31) = 51.2$, $p < 0.001$).

Differences in response time across HMM-inferred states suggested that these labels produced meaningful differences in primary behavioral metrics. To eliminate the possibility that exploration was merely disengagement from the task, we examined average reward retrieval time during exploratory and exploitative states. There was no significant difference in reward retrieval time between two states (*Figure 1—figure supplement 1G*, t-test, $t(31) = 0.05$, $p = 0.95$), suggesting that animals were not more disengaged from the task during exploration than exploitation. They were only slower in making a decision. Together, these results demonstrated that HMM-labeled exploration was meaningful, non-reward-maximizing, and accompanied by enhanced reward learning, matching the normative definition of exploration.

## Males made more exploratory choices than females because they explored for longer periods of time once they started

With more confidence in the validity of HMM-inferred states, we found that males, on average, were more likely to be in the exploratory regime than the exploitative one, with 72.9% ± 11.5% STD of trials labeled as exploratory (*Figure 1E*). Females explored much less with only 55.4% ± 20.4% STD of trials being exploratory (*Figure 1F*, t-test, $t(30) = 2.98$, $p = 0.0056$; 95% CI for the difference between the sexes = [5.5%, 29.3%]). As groups, males and females were reasonably, but not perfectly discriminable in terms of the proportion of exploratory choices (*Figure 1G*, receiver operating characteristic analysis, AUC = 0.76 ± 0.09, 95% CI for the difference = [0.59, 0.93], $p = 0.013$). These differences were largely driven by the greater male tendency to keep exploring once they started. Males repeated exploration 92.1 % ( ± 3.4 % STD) of the time, while females stopped exploring and committed to a choice more quickly, repeating exploration only 83.1 % ( ± 16.8 % STD) of the time (t-test, $t(30) = 2.09$, $P = 0.045$, 95% CI for the difference = [0.2%, 17.8%]). There were no significant differences in the other parameter of the HMM (probability of repeating exploitation: males = 83.5% ± 3.7%; females = 79.7 ± 22.1 %; $t(30) = 0.69$, $p = 0.5$). Since males had more exploratory trials, which took longer, we tested the possibility that the sex difference in response time was due to prolonged exploration in male by calculating a two-way ANOVA between explore-exploit state and sex in predicting response

time. There was a significant main effect of state (main effect of state: F (1,30) = 13.07, p = 0.0011), but males were slower during females during both exploitation and exploration (main effect of sex, F(1,30) = 14.15, p = 0.0007) and there was no significant interaction (F (1,30) = 0.279, p = 0.60). We also examine whether the probability of exploration changed over trials or across sessions by calculating the probability of exploration early, mid, and late within one session and across sessions. However, we failed to see changes in the amount of exploration within sessions and across sessions in both males and females.

Although sex differences in model parameters were modest, analyzing the full dynamics of the fitted HMMs again supported a robust sex difference in the tendency to explore (see Materials and methods). In models fit to males, exploration was a deeper, more 'sticky' behavioral state (*Figure 1H*, stationary probability of exploration = 68.0% ± 8.5% STD), compared to models fit to females, where exploration and exploitation were more closely matched (54.4% ± 18.4% STD; different from males: t(30) = 2.68, p = 0.012, 95% CI for the difference = [3.2%, 23.9%]). This suggests that males were more likely to get 'stuck' in an extended exploratory period, requiring more energy to escape from exploring and start exploiting a good choice.

## Multiple variables in reinforcement learning models may be the cause of increased exploration

The results from the HMM analyses suggest that males were, on average, more exploratory than females, and not because they were more likely to initiate exploration, but because they were more likely to become 'stuck' in exploration. This suggests that there were sex differences in the animals' approach to this task. However, a crucial question remained unanswered: what computational differences made the males more exploratory? To address this question, we turned to reinforcement learning (RL) modeling to look for individual variability in latent cognitive parameters that could influence exploration and exploitation (*Daw et al., 2006*; *Ishii et al., 2002*; *Jepma and Nieuwenhuis, 2011*; *Pearson et al., 2009*).

RL models include multiple parameters that could influence exploration. Consider a simple, two-parameter RL model, with one learning rate parameter (α) and one parameter for decision noise (inverse temperature β). Traditionally, only the latter parameter is thought to be related to exploration, and many previous studies of exploration have focused exclusively on this inverse temperature parameter (*Beeler et al., 2010*; *Cinotti et al., 2019*). However, exploration in an RL model should be a function of both the difference in subjective values and the decision-noise in the model. This is because both parameters increase the likelihood that agents will make non-reward-maximizing decisions (*Figure 2A*). In the case of decision-noise, this happens because more choices deviate from reward-maximizing policies as noise increases. In the case of learning rate, this happens because the learning rate controls how quickly agents can move away from the regime in which decision-noise is highest. To test our intuition, we simulated data from a simple two-parameter RL model, then used the HMM to infer when and why exploration occurred. As expected, changing the decision-noise parameter (β) robustly changed the probability of exploration (*Figure 2B*; GLM, main effect of inverse temperature, β1 = –0.11, p < 0.0001): the larger the inverse temperature, the lower the decision noise and the lower probability of exploration. Critically, learning rate (α) also influenced the probability of exploration (*Figure 2B*; GLM, main effect of learning rate, β2 = –0.10, p < 0.0001). In fact, there were some values for learning rate (α) at which changing decision noise had no effect on exploration whatsoever: α and β interacted to influence exploration (*Figure 2C*; GLM, interaction term, β3 = –0.38, p < 0.0001). This occurred because when the learning rate was very low, agents failed to move away from the high-decision-noise regime at all, meaning that there was little effect of any additional noise. Thus, even in this simple two-parameter RL model, multiple latent, cognitive variables can influence exploration. Because of this ambiguity, it was not clear whether males explored more frequently because they had more decision-noise, because they learned less from rewards, or because there were changes in other decision-making or learning computations, like the tendency to simply repeat past choices. Fortunately, we can distinguish these possibilities via fitting RL models to the data.

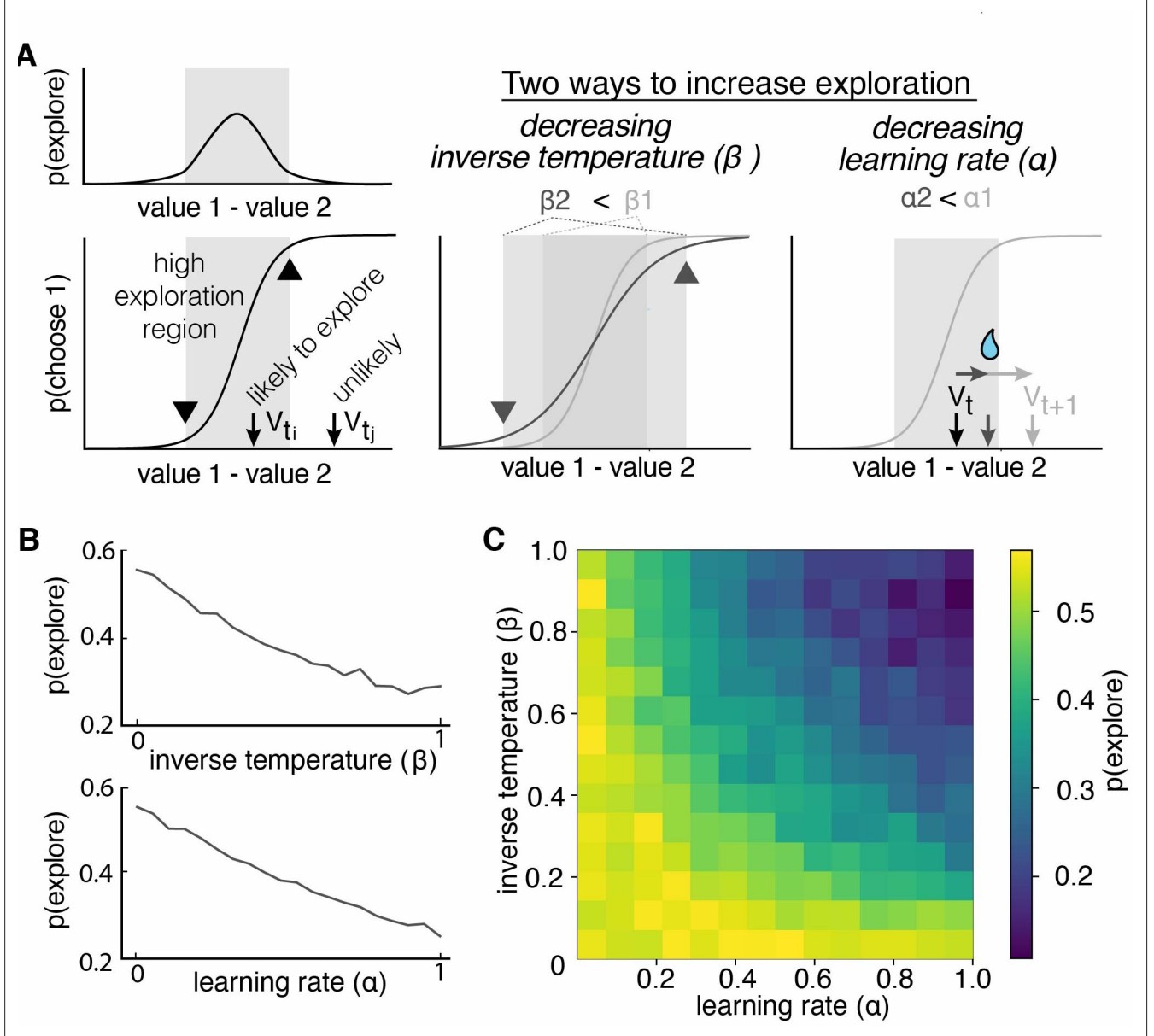

**Figure 2.** Multiple reinforcement learning parameters can influence the probability of exploration. (**A**) Exploration occurs most often when option values are close together, illustrated by the gray shaded boxes in the value-choice functions. Both decreasing inverse temperature (β) and decreasing learning rate increases exploration because each manipulation changes the amount of time spent in the high exploration zone, although the mechanisms are different. Decreasing inverse temperature (β) widens the zone by flattening the value-choice function and increasing decision noise. Decreasing learning rates (α) keeps learners in the zone for longer. (**B**) Probability of exploration from 10,000 different reinforcement learning agents performing this task, initialized at different random combinations of inverse temperatures (β) and learning rates (α). Marginal relationships between decision noise (top) and learning rate (bottom) are shown here. (**C**) Heatmap of all pairwise combinations of learning rate and inverse temperature.

## Sex differences in exploration was due to changes in the learning rate - females had higher learning rates than males

The HMM suggested that males and females had different levels of exploration, but it did not provide insight into the latent, cognitive processes behind these differences. Fortunately, RL models allow us to identify differences in a variety of latent cognitive variables that could influence exploration, either alone or in combination. However, our ability to make inferences about changes in model parameters is highly sensitive to the correct specification of the model, so we first had to identify the best-fitting RL model for these animals.

There are many ways to parameterize RL models (*Katahira, 2018*), the majority of which can be put in three categories: value-dependent learning terms, value-independent bias terms, and decision noise/randomness terms. Previous studies have shown the effect of various RL parameters on decision making, including learning terms such as asymmetrical learning rate (*Frank et al., 2007*; *Gershman, 2016*), bias terms such as choice bias (*Katahira, 2018*; *Wilson and Collins, 2019*), noise terms such as inverse temperature and lapse rate (*Economides et al., 2015*; *Wilson and Collins, 2019*). Here, we compared seven reinforcement learning models that made different assumptions about the latent processes mice might be employing via different combinations of learning, bias, and noise terms. These models included: (1) a 'random' model with some overall bias for one choice over the other, (2) a 'noisy win stay lose shift' model that assumes a win stay lose shift policy with some level of randomness, (3) a two-parameter 'RL' model with a consistent learning rate and some inverse temperature that captures decision noise, (4) a three-parameter 'RLε' model with a consistent learning rate, and inverse temperature that captures value-based decision noise, and a value-independent noise, (5) a four-parameter 'RLCK' model that captures both value-based and value-independent decision with separate parameters for learning rate, decision noise, choice bias, and choice stickiness, (6) a five-parameter 'RLCKγ' model that incorporates differential learning rate for rewarded and unrewarded outcomes on top of the 'RLCK' model, (7) a five-parameter 'RLCK $\eta$ ' model that adds a parameter that tunes the weight between value-based and choice-based decision to the 'RLCK' model (see Materials and methods, *Figure 3A*).

Although model fitting was slightly different across sexes, in both males and females, the "RLCK" model, four-parameter model with value and choice kernel updating policies, best characterized animals' choice behaviors in this task among all seven models (*Figure 3B*). The fact that the 'RLCK' model was the best-fit model in both males and females does not mean both sexes had the same strategy or that RL modeling cannot capture those strategies. Instead, this may suggest that strategic differences between sexes may be more a matter of degree (i.e.: differences in the specific values of model parameters), rather than a matter of categorically distinct computation. This interpretation also makes the most sense in light of the biology of sex differences, which produce few (if any) truly categorically distinct changes in neural function, but rather serve to bias neural systems across sexes in multiple complex ways.

To quantify how well each RL model was at predicting animals' choices, we measured the model agreement for each model, which was calculated as the probability of choice correctly predicted by the optimized model parameters for each model (*Figure 3C*). Then we conducted a multiple comparison across model agreement of RL models (test statistics reported in *Supplementary file 2*). The results suggested that the RL models with parameter(s) that account for choice bias (RLCK, RLCKγ, RLCK $\eta$ ) were significantly better at predicting animals' actual choices than the models that do not account for choice bias and non-RL models (random, noisy WSLS, RL, RLε). There was no significant difference in model agreement between RLCK, RLCKγ, and RLCK $\eta$ . Based on the result of model comparison (AIC) and model agreement, we decided that the four-parameter RLCK model is the simplest, best-fit model that best predicted animal' actual choices. Finally, to visualize how well the RLCK model was at predicting choices of animals with different learning performance, we plotted the simulated choices and actual choices against the matching law (*Poling et al., 2017*), which dictates that the probability of choice is proportional to the probability of reward. The figure showed that this four-parameter model was able to characterize animals' choice behaviors regardless of the value-based learning performance (*Figure 3—figure supplement 1*).

The RLCK model had four parameters, which we then compared across sexes. We found that females had significantly higher learning rate (α) than males (*Figure 3D*, $t(30) = 2.40$, $p = 0.02$) but there was no significant difference across sexes in other parameters (β: $t(30) = 1.44$, $p = 0.16$; $\alpha_c$: $t(30) = 1.40$, $P = 0.17$; $\beta_c$: $t(30) = 1.73$, $p = 0.09$). To examine whether the higher learning rate in females was driven by a few individuals with extremely high learning rates, we plotted the distribution and calculated the separability of learning rates of two sexes. As groups, males and females were reasonably discriminable in terms of the learning rate (*Figure 3E*, receiver operating characteristic analysis, AUC = $0.72 \pm 0.09$, 95% CI for the difference = [0.54, 0.90], $p = 0.035$). These results suggested that the difference in the level of exploration between males and females was not due to differences in decision noise, but instead due to differences in learning rate.

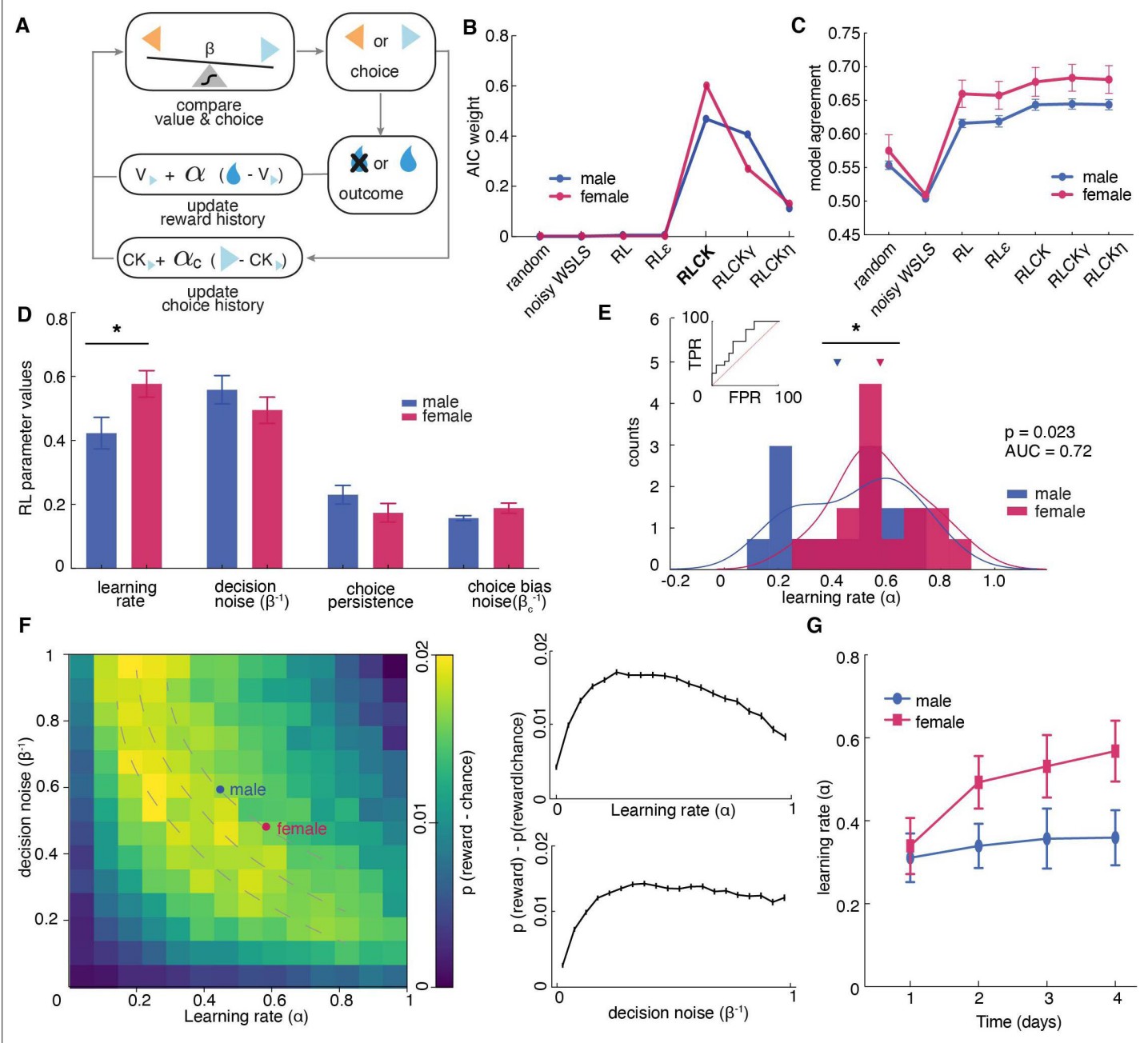

**Figure 3.** Sex differences in learning rate, but not decision noise, drove differences in explore-exploit decisions. (**A**) A diagram of latent parameters that capture learning ($\alpha$), bias ($\alpha_c$), inverse temperature ($\beta$) in reinforcement learning models. The models tested used a combination of these parameters (see Materials and methods). (**B**) Model comparison across seven reinforcement learning models with various parameter combinations for males and females. The four-parameter reinforcement learning-choice kernel (RLCK) model has the highest relative likelihood in both sexes. (**C**) Model agreement across seven reinforcement learning models, which measures how well a model predicts the actual choices of animals. (**D**) All four parameters in the best fit RLCK model across sexes. Learning rate ($\alpha$) was significantly higher in females than males. (**E**) Distribution of learning rate across sexes. (**F**) (left) Simulation of reward acquisition of RL agent with different combinations of learning rate ($\alpha$) and decision noise ($\beta^{-1}$). Different combinations of learning rate and decision noise can result in the same level of reward performance. Average learning rate and decision noise is overlaid on the heatmap for males and females. (right) relationship between reward acquisition and learning rate or decision noise separately. High learning rate is not equivalent to better learning performance. (**G**) Learning rate in females increased across days, suggestive of meta learning. * indicates $p < 0.05$. Graphs depict mean ± SEM across animals.

The online version of this article includes the following figure supplement(s) for figure 3:

**Figure supplement 1.** The best fit model, the four-parameter reinforcement learning-choice kernel (RLCK) model, captured both value-dependent and value-independent choice behaviors.

While females had significantly higher learning rate (α) than males, they did not obtain more rewards than males. This is because the learning rate parameter in an RL model does not equate to the learning performance, which is better measured by the number of rewards obtained. The learning rate parameter reflects the rate of value updating from past outcomes. Performing well in this task requires both the ability to learn new information and the ability to hang onto the previously learned information. That occurs when the learning rate is moderate but not maximal. When the learning rate is maximal ($\alpha = 1$), only the outcome of the immediate past trial is taken into account for the current choice. This essentially reduces the strategy to a win-stay lose-shift strategy, where choice is fully dependent on the previous outcome. A higher learning rate in a RL model does not translate to better reward acquisition performance. To illustrate that different combinations of learning rate and decision noise can result in the same reward acquisition performance. We conducted computer simulations of 10,000 RL agents defined by different combinations of learning rate (α) and inverse temperature ($\beta^{-1}$) and plotted their reward acquisition performance for the restless bandit task (*Figure 3F*, temperature instead of inverse temperature was plotted for the ease of presentation). This figure demonstrates that (1) different learning rate and inverse temperature combinations can result in similar performance, (2) the optimal reward acquisition is achieved when learning rate is moderate. This result suggested that not only did males and females had identical performance, their optimized RL parameters put them both within the same predicted performance gradient in this plot.

One interesting finding is that, when compared learning rate across sessions within sex, females, but not males, showed increased learning rate over experience with task (*Figure 3G*, repeated measures ANOVA, female: main effect of time, F (2.26,33.97) = 5.27, p = 0.008; male: main effect of time, F(2.5,37.52) = 0.23, p = 0.84). This points to potential sex differences in meta-learning that could contribute to the differential strategies across sexes.

## Females learned more during exploratory choices than males

The results of HMM model and RL models revealed significant sex differences in exploration, paralleled by sex differences in rate of learning. What remains unclear is how sex, explore-exploit states, and reward outcomes all interact together to influence the animals' choices. Therefore, we conducted a four-way repeated measures ANOVA to examine how (1) positive and negative outcomes, (2) explore-exploit states, (3) sex, and (4) subject identity (nested in sex) all came together to influence choice: whether animals would repeat their last choice (stay) or try a different option (switch; *Supplementary file 1*). This four-way repeated measure ANOVA allowed us to understand the main effect of sex, state, and outcome, as well as all pair and triplet-wise interaction effects, on how animals learned from previous rewards. Note that in previous analyses, we used subject averaged data but since subject average (16 subjects each sex) is underpowered to detect a three-way interaction effect, we used session averaged data to increase the power to detect any effects across sex, state, and outcome.

The results revealed an expected significant main effect of outcome on stay-switch decisions (main effect of outcome, p < 0.00001). This effect was driven by the tendency of animals to repeat the previous choice (i.e.: not switching) after obtaining a reward (post hoc t-test compared to chance at 0.50, mean = 0.75, 95% CI = [0.74, 0.77], t(255)=34.33, p < 0.0001) and a much smaller tendency to switch more often than chance after reward omission (mean = 0.52, 95% CI = [0.50, 0.55], post hoc t-test, t(255) = 2.05, p = 0.04). The tendency to switch or stay also differed by sex, with females more likely to repeat a previous choice and males more likely to switch (main effect of sex, p < 0.00001; post-hoc t-test on p(switch): t(254) = 4.12, p < 0.0001). There was also a significant interaction effect between sex and outcome (sex X reward interaction, p < 0.00001). To understand how reward and reward omission differentially affect choice across sexes, we conducted post-hoc win-stay lose-shift analyses. We found that female mice displayed more win-stay behaviors, indicating that they were more likely than the males to repeat behaviors that produced reward on the previous trial (*Figure 4A*, sex X reward interaction, p < 0.000001, *Supplementary file 1*; post hoc t-test: t(254) = 5.53, p < 0.0001). As groups, males and females were reasonably, but not perfectly discriminable in terms of the proportion of win stay choices (*Figure 4B*, receiver operating characteristic analysis, AUC = 0.74 ± 0.09, 95% CI for the difference = [0.56, 0.92], p = 0.0195). In contrast, male mice tended to shift even when the previous choice was rewarded.

There was no significant sex difference in learning from losses (*Figure 4C* and t(25) = 1.40, p = 0.16), but this did not mean that sex differences in learning were solely due to sex differences in

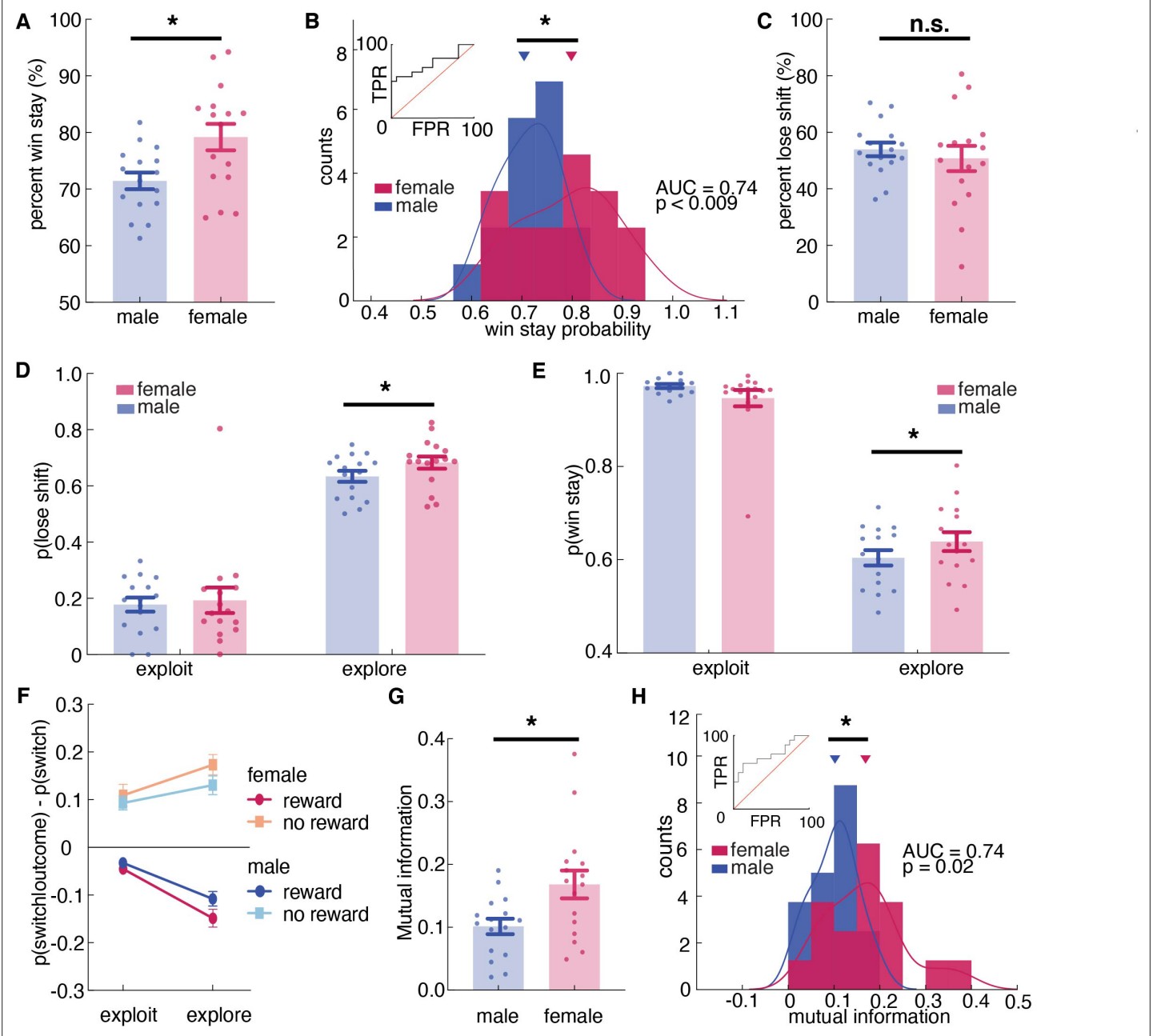

**Figure 4.** Females used more information about past outcomes and past choices to make decisions, and learned more during exploration. (**A, B**) Percent win stay behavior (A: average; B: distribution) reveals that females were more likely to stay with the same choice after a reward. (**C**) Average percentage of lose shift behavior across sexes. (**D**) Probability of shifting after a loss during explore or exploit trials. (**E**) Probability of staying after a win during explore or exploit trials. (**F**) The probability of males and females switching targets on the next trial, given the current trial's outcome and latent state. Females learned more only during exploratory trials. (**G, H**) Average (**G**) and distribution (**H**) of percentage of mutual information across all trials in females and males reveals that females use more information about past trials (choice and outcome) in making future decisions. * indicates p < 0.05. Graphs depict mean ± SEM across animals.

The online version of this article includes the following figure supplement(s) for figure 4:

**Figure supplement 1.** Reward learning in explore vs. exploit state across sexes.

learning from wins. There were at least two ways that we could observe an equivalent tendency to lose-shift across sexes in this task. One possibility is that females only learn more from positive outcomes, but not negative ones. However, the other possibility is that this lack of a difference in lose-shift behaviors between males and females was an artifact of the tendency of males to explore

more frequently. Since males spend more time exploring (*Figure 1E*) and learning from both wins and losses is enhanced during exploration (*Figure 1—figure supplement 2F*), males could lose-shift less frequently than females during both exploration and exploitation, yet still lose-shift exactly as much as females because a greater proportion of their choices were exploratory.

To dissociate these possibilities, we next examined the effects of exploration and exploitation. The ANOVA revealed a significant main effect of state, resonating with the result of HMM model validation that animals were more likely to switch during exploratory state than during exploitative state (*Supplementary file 1*, main effect of state, p < 0.00001; sex X state interaction, p = 0.0667; *Figure 1—figure supplement 2F*). Critically, there was also a significant three-way interaction between sex, explore/exploit state, and reward (*Figure 4F*, *Figure 4—figure supplement 1*, *Supplementary file 1*, sex X reward X state interaction, p = 0.0438). This could imply sex-linked differences in reward learning across exploration and exploitation. To determine if this was true, we separated out the probability of lose-shift according to whether it happened during exploration or exploitation, as inferred from the Hidden Markov model. Males switched less after losses than females within exploratory states (*Figure 4D*; post hoc t-test on session averages: sex difference within exploration: p < 0.001, t(251) = 3.39), though there was no significant sex difference within exploitation (p = 0.06, t(243) = 1.87; note that differing degrees of freedom are due to the fact that exploitation was not observed in some sessions for some animals). This supported the second hypothesis that males lose-shift less than females both when exploring and when exploiting, but that there was no difference in lose-shift overall because males spent more time in a state in which both win-stay and lose-shift choices occur more frequently (*Figure 1—figure supplement 2F*). We also found that the increased tendency to win-stay that we observed in females was driven by the explore choices (*Figure 4E*; post-hoc t-test: p = 0.015, t(251) = 2.55). There was no significant difference in win-stay between males and females during exploit choices (post-hoc t-test: p = 0.09, t(244) = 1.68). Together these results suggest that females were better explorers (i.e.: they had increased reward learning during exploration), whereas males learned slower during exploration but compensated for this learning disadvantage by exploring more frequently.

These effects were not driven by idiosyncratic strategic differences between the sexes (e.g. shifting only after two losses). We used a model-free approach to quantify the extent to which behavior was structured without making strong assumptions about what form this structure might take. We calculated conditioned mutual information for all sessions across sexes (*Leao et al., 2004*; *Wyner, 1978*), to examine how choice behavior was influenced by information of past choice history, given the immediate outcome. The result suggested that mutual information was higher in females than males, suggesting that females were using more information from the past choice and outcome to make their current decision. (*Figure 4G*, t(30) = 2.65, p = 0.013). As groups, males and females were reasonably discriminable in terms of mutual information (*Figure 4H*, receiver operating characteristic analysis, AUC = 0.74 ± 0.09, 95% CI for the difference = [0.57, 0.92], p = 0.0195). Together, these results reinforced our conclusion that females were learning more: utilizing more information from the past trial to make current choices.

## Discussion

Sex mechanisms biasing the preferred approaches taken during cognitive tasks are potentially significant contributors to task performance. We used a combination of computational modeling approaches to characterize sex differences in a canonical explore/exploit task. While males and females had similar performance, they used different latent explore-exploit strategies to learn about the dynamic environment. Males explored more than females and were more likely to 'get stuck' in an extended period of exploration before committing to a favored choice. In contrast, females were more likely to show elevated learning during the exploratory period, making exploration more efficient and allowing them to start exploiting a favored choice earlier. Furthermore, the learning rate increased over days in females but stayed stable in males. Such meta-learning in females permitted learning about the current task (which option provides the best reward outcome), as well as the structure of the task (the reward probability of choices changes over time). This allowed them to shift more quickly to exploit a rewarding option when they found one and only explored when the current option failed to provide valuable rewards. Together, these results demonstrate that while the overall performance was similar, males and females tended to adopt different strategies for interacting

with the same uncertain environment. The difference in explore-exploit strategies across sexes may provide us insight into potential sex-modulated mechanisms that are implicated in learning under uncertainty.

Our major finding that males learned less during exploration and explored for longer is consistent with two explanations. First, males learned more slowly during exploration and as a result, they had to explore for longer to learn which option was worth exploiting. Another possibility is that males got 'stuck' in extended periods of exploration that prohibited them from applying the knowledge they have learned and committing to a rewarding option. In this view, it takes some energy to stop exploring and transition to exploit. As we have shown in our result, males had 'deeper' exploratory states and they were more likely to keep exploring once started. These two explanations are not mutually exclusive because changes in learning could contribute to changes in the stickiness of exploration and vice versa. This essentially presents a chicken and egg problem. It is difficult to distinguish from behavior alone whether slower learning drives longer exploration, or vice versa, if being stuck in exploration results in slower learning. Neural measures during explore and exploit choices across sexes may help us differentiate learning signals from signals that drive exploration, and whether these signals are sex-different.

Answering these neural questions will require a way to reliably identify latent exploration and exploitation states. The Hidden Markov model (HMM) has been used to infer trial-by-trial exploration and exploitation in non-human primates (*Ebitz et al., 2018*). The HMM inferred latent goal states explained more variance in neural activity than other decision-variables (*Ebitz et al., 2018*). In our data, we have shown for the first time that the HMM model was able to label a meaningful exploratory state that matches normative definitions of exploration in the mouse model. In the future, this computational tool can complement neurobiological recording techniques to examine for neural correlates of exploration on a trial-by-trial basis, and permit the visualization of dynamic landscapes of choice behavior across individuals (as in *Figure 1H*) or with pharmacological or other challenges (*Ebitz et al., 2019*).

Reinforcement learning (RL) models have also been used in the past to identify levels of exploration across individuals (*Daw et al., 2006*; *Pearson et al., 2009*). However, our findings indicate that multiple latent parameters can influence how much these models explore. Here, we found that differences in exploration between the sexes were due to differences in learning rates, not due to differences in the decision noise parameter, which is more commonly associated with exploration. While RL models are helpful for understanding cognitive or computational mechanisms, they are limited in their ability to identify when exploration is happening. The HMM model, conversely, provides no insight into mechanisms, but can tell us precisely when exploration is occurring, both in animal behaviors and RL models (*Ebitz et al., 2018*). By combining the HMM and RL approaches, we capitalized on the advantages of both frameworks: linking changes in exploration across sexes to underlying mechanisms of exploration. Since the broader reinforcement learning model framework is highly adaptive and amenable, in the future, the HMM and kinds of model-free analyses we completed here could also inform the design of RL models to capture explore-exploit decisions more precisely.

Future work is needed to understand the neurobiological bases of these observations. One neuromodulator that is implicated in reinforcement learning, including the transition between exploration and exploitation, *and* is strongly sex-modulated is dopamine (*Beeler et al., 2010*; *Jenni et al., 2017*). Studies have shown that dopamine signals regulate exploration via mechanisms of action selection and learning (*Beeler et al., 2010*; *Frank et al., 2009*; *Humphries et al., 2012*). However, due to the exclusive use of males in many foundational experiments (*Beeler et al., 2010*; *Cinotti et al., 2019*), the fact that dopamine function on decision making is strongly modulated by sex and sex-linked mechanisms is often overlooked. For example, estradiol has been demonstrated to exert both acute and chronic modulatory effects on dopamine release, reuptake, and receptor binding (*Yoest et al., 2014*), allowing enhanced DA release and faster DA clearance in females (*Becker, 1999*). This mechanism could contribute to increased reward learning observed in females during exploration. The prefrontal cortex (PFC), which receives dopamine projections, is a target brain region to understand exploration and exploitation (*Ebitz et al., 2018*; *Jenni et al., 2017*). Our previous study implicated PFC in the differences in learning strategy between males and females (*Chen et al., 2021b*). It is possible that prefrontal cortical dopamine is particularly engaged in implementing explore-exploit strategies via sex-biased mechanisms of learning.

Rodent operant testing is frequently used to assess cognitive functions. This is critical for translational work in animals needed to link pharmacology, genetics, and other potential biological contributors to behavior (*Grissom and Reyes, 2019*). However, many classic rodent cognitive tasks are species-specific: they were not designed to assess the same cognitive processes across species, and this limits their translational ability. Currently, there is an emerging exciting trend among rodent researchers with adopting tasks with translational potentials in rodents, such as reversal learning and various versions of bandit task (*Izquierdo et al., 2019*; *Groman et al., 2016*; *Bari et al., 2019*; *Grossman et al., 2021*), where we can assess the same cognitive processes across species and across animal models of diseases. Here, we establish that explore-exploit decisions are shared cognitive processes across species, making this restless bandit task an ideal tool for translational research. Parallel approaches in humans have been used to examine the explore-exploit strategic phenotype of neuropsychiatric disorders, including ADHD, addiction, and depression (*Addicott et al., 2021*; *Addicott et al., 2012*; *Beeler et al., 2012*; *Blanco et al., 2013*). The computational modeling used here permits fine grained quantification of individual variability in latent parameters that capture adaptive changes in exploration in changing environments. The computational approaches we develop here could help identify behavioral endophenotypes across species underlying a variety of neuropsychiatric disorders and open up new avenues for understanding as well as rescuing dysfunction in value-based decision making.

## Materials and methods

### Key resources table

| Reagent type (species) or resource | Designation | Source or reference | Identifiers | Additional information |
|---|---|---|---|---|
| Strain, strain background (mouse) | B6129SF1/J | The Jackson Laboratory | JAX: 101,043 | |
| Software, algorithm | Python 3 | Python | SCR_008394 | |
| Software, algorithm | MATLAB R2013a | MathWorks | SCR_001622 | |

### Animals

Thirty-two BL6129SF1/J mice (16 males and 16 females) were obtained from Jackson Laboratories (stock #101043). Mice arrived at the lab at 7 weeks of age, and they were housed in groups of four with ad libitum access to water while being mildly food restricted (85–95% of free feeding weight) for the experiment. Animals engaging in operant testing were housed in a 0900–2100 hours reversed light cycle to permit testing during the dark period. Before operant chamber training, animals were food restricted to 85–90% of free feeding body weight. Operant testing occurred five days per week (Monday-Friday). All animals were cared for according to the guidelines of the National Institution of Health and the University of Minnesota.

### Apparatus

Sixteen identical triangular touchscreen operant chambers (Lafayette Instrument Co., Lafayette, IN) were used for training and testing. Two walls were black acrylic plastic. The third wall housed the touchscreen and was positioned directly opposite the magazine. The magazine provided liquid reinforcer (Ensure) delivered by a peristaltic pump, typically 7 µl (280 ms pump duration). ABET-II software (Lafayette Instrument Co., Lafayette, IN) was used to program operant schedules and to analyze all data from training and testing.

### Behavioral task

#### Two-armed spatial restless bandit task

Animals were trained to perform a two-armed spatial restless bandit task in the touchscreen operant chamber. Each trial, animals were presented with two identical squares on the left and right side of the screen. Nose poke to one of the target locations on the touchscreen was required to register a response. Each location is associated with some probability of reward, which changes independently over time. For every trial, there is a 10 % chance that the reward probability of a given arm will increase or decrease by 10 %. All the walks were generated randomly with a few criteria: (1) the overall reward probabilities of two arms are within 2 % of each other, preventing one arm being overly better than

the other, (2) the reward probability cannot go down to 0 % or go up to 100%, (3) there are no 30 consecutive trials where the reward probabilities of both arms are lower than 20 % to ensure motivation. Animals ran a simple deterministic schedule on Monday to re-adapt to operant chamber after weekends off and ran a different restless bandit task each day from Tuesday to Friday. Animals ran for two rounds of four consecutive days and within each day, animals completed either 300 trials or spent a maximum of two hours in the operant chamber. On average across all sessions, animals performed 276.5 trials with a standard deviation of 8.6 trials (male average: 253.7 trials, sd = 15.4; female average 299.3 trials, sd = 0.74). Data was recorded by the ABET II system and was exported for further analysis. All computational modeling was conducted using python. All behavioral data have been deposited in generic database (Dyrad) with accession link https://doi.org/10.5061/dryad.z612jm6c0. Codes used can be found at https://github.com/CathySChen/restlessBandit2021 (*Chen, 2021a* copy archived at swh:1:rev:a0a377707627f93f3637e1520f9d1304121dcf1a).

## Data analysis

### General analysis techniques

Data was analyzed with custom PYTHON, MATLAB, and Prism eight scripts. Generalized linear models, ANOVA, and t-test were used to determine sex differences over time, unless otherwise specified. p Values were compared against the standard $\alpha = 0.05$ threshold. The sample size is n = 16 for both males and females for all statistical tests. No animal was excluded from the experiment. One outlier was removed in one analysis using ROUT method (with Q set to 1%). This outlier was from the animal that ran the lowest number of total trials. Statistics for both no outlier removal and outlier removal were reported in the result. All statistical tests used and statistical details were reported in the results. All figures depict mean ± SEM.

### Mixture model

We first asked whether there were different behavioral dynamics that might correspond to exploration and exploitation. Exploration and exploitation take place on different time scales. In RL agents, for example, exploration is typically implemented via adding noise or indeterminacy to a decision-rule. The identity of choices that are caused by this noise—the exploratory choices—will thus switch more frequently than the identity of choices that depend on option value. We should see short runs of exploratory choices and long runs of exploitative ones (*Ebitz et al., 2018*). To the extent that choice runs end probabilistically (an assumption of the HMM framework), choice run durations (inter-switch intervals) will be exponentially distributed (*Berg, 1993*). Since there exist multiple causal regimes (such as exploration and exploitation), inter-switch intervals will be distributed as a mixture of multiple exponential distributions (*Figure 1—figure supplement 2A*). Because trials are discrete, rather than continuous, we fit mixtures of the discrete equivalent to the exponential distribution, the geometric distribution. We examined the distribution of 24,836 interswitch intervals. Adding a second mixing distribution significantly improved model fit (one-component, one-parameter mixture log-likelihood: –44555, two-component, three-parameter: –41656; likelihood ratio test, $p < 10^{-32}$). Adding additional mixing distributions continued to improve model fit, a common observation in mixture modeling. However, the continued improvement was substantially less than the leap from one to two components (*Figure 1—figure supplement 2A*; three-component: –41431, four-component: –41412) and additional mixtures beyond two had weights below 3 %. This suggests that a mixture of one fast-switching regime and one slow-switching regime was the most parsimonious explanation for the data (*McLachlan and Peel, 2004*).

### Hidden Markov model (HMM)

In order to identify when animals were exploring or exploiting, we fit an HMM. In an HMM framework, choices (y) are 'emissions' that are generated by an unobserved decision process that is in some latent, hidden state (z). Latent states are defined by both the probability of making each choice (k, out of $N_k$ possible options), and by the probability of transitioning from each state to every other state. Our model consisted of two types of states, the explore state and the exploit state. The emissions model for the explore state was uniform across the options. The emissions model for the explore state was uniform across the options:

$$p\left(y_t = k | z_t = explore\right) = \frac{1}{N_k}$$

This is simply the maximum entropy distribution for a categorical variable - the distribution that makes the fewest number of assumptions about the true distribution and thus does not bias the model towards or away from any particular type of high-entropy choice period. This does not require, imply, impose, or exclude that decision-making happening under exploration is random. *Ebitz et al., 2019* have shown that exploration was highly structured and information-maximizing, despite being modeled as a uniform distribution over choices (*Ebitz et al., 2020*; *Ebitz et al., 2019*). Because exploitation involves repeated sampling of each option, exploit states only permitted choice emissions that matched one option. That is:

$$\begin{cases} p\left(y_t = k | z_t = exploit_i, k \in exploit_i\right) = 1 \\ p\left(y_t = k | z_t = exploit_i, k \notin exploit_i\right) = 0 \end{cases}$$

The latent states in this model are Markovian, meaning that they are time-independent. They depend only on the most recent state ($z_t$):

$$p\left(z_t | z_{t-1}, y_{t-1}, \ldots, z_1, y_1\right) = p\left(z_t \vee z_{t-1}\right)$$

This means that we can describe the entire pattern of dynamics in terms of a single transition matrix. This matrix is a system of stochastic equations describing the one-time-step probability of transitioning between every combination of past and future states (i, j).

$$p\left(z_t = i \vee z_{t-1} = j\right)$$

Here, there were three possible states (two exploit states and one explore state) but parameters were tied across exploit states such that each exploit state had the same probability of beginning (from exploring) and of sustaining itself. Transitions out of the exploration, into exploitative states, were similarly tied. The model also assumed that the mice had to pass through exploration in order to start exploiting a new option, even if only for a single trial. This is because the utility of exploration is to maximize information about the environment, as defined in both animal foraging literature and reinforcement learning models (*Mehlhorn et al., 2015*). If an animal switches from a bout of exploiting one option to another option, that very first trial after switching should be exploratory because the outcome or reward contingency of that new option is unknown and that behavior of switching aims to gain information. Through fixing the emissions model, constraining the structure of the transmission matrix, and tying the parameters, the final HMM had only two free parameters: one corresponding to the probability of exploring, given exploration on the last trial, and one corresponding to the probability of exploiting, given exploitation on the last trial.

The model was fit via expectation-maximization using the Baum Welch algorithm (*Bilmes, 1998*). This algorithm finds a (possibly local) maxima of the complete-data likelihood. A complete set of parameters θ includes the emission and transition models, discussed already, but also initial distribution over states, typically denoted as PI. Because the mice had no knowledge of the environment at the first trial of the session, we assumed they began by exploring, rather than adding another parameter to the model here. The algorithm was reinitialized with random seeds 20 times, and the model that maximized the observed (incomplete) data log likelihood across all the sessions for each animal was ultimately taken as the best. To decode latent states from choices, we used the Viterbi algorithm to discover the most probable a posteriori sequence of latent states.

To account for the effect of reward on choice dynamics, we extended the two-parameter HMM model to an input-output HMM model (4-parameter ioHMM), whose framework allows inputs, such as reward outcomes, to influence the probability of transitioning between states (Bengio and Frasconi; *Ebitz et al., 2019*). The ioHMM model improved model fit (two-parameter original HMM: log-likelihood = −1424.0; 4-parameter ioHMM: log-likelihood = −1430.5). Typically, improved model fit is expected with the addition of parameters. To determine whether it's a meaningful improvement of model fit that justifies doubling the number of parameters, we calculated AIC and BIC for model comparison. The result of model comparison using both AIC suggested that the original two-parameter model was the better model (AIC: two-parameter original HMM: AIC = 2976.1; four-parameter ioHMM: AIC

= 3117; relative likelihood (AIC weight) of the four-parameter ioHMM <10^–30). BIC test has a even larger penalty for more parameters and therefore selected against the four-parameter ioHMM (BIC: two-parameter original HMM: BIC = 3562.8; two-parameter ioHMM: BIC = 4290.4; relative likelihood (BIC weight) of the four-parameter ioHMM <10^–150). While reward outcomes could affect the probability of state transitions, the model comparison suggested that the extra input layer did not explain more variance in choice dynamics, and therefore, we would favor the original, simpler two-parameter HMM model.

To account for the effect of biased exploitation on the probability of transitioning between states, we also considered an unrestricted HMM model with no parameter tying (four parameter ntHMM), where we treat exploiting the left side and exploiting the right side as two separate exploit states and allow differential transition probability to each exploit state. However, the ntHMM did not improve model fit with almost identical log-likelihood (two-parameter original HMM: log-likelihood = –1424.0; four-parameter ntHMM: log-likelihood = –1423.8). Then we compared two models by calculating the AIC/BIC values for both models, which penalized extra parameters in the no parameter-tying HMM. The AIC test favored the original two-parameter model (AIC: two-parameter original HMM: AIC = 2976.1; four-parameter ntHMM: AIC = 3103.5; relative likelihood (AIC weight) of the four-parameter ntHMM <10^–28). The BIC test also favored the simpler two-parameter model (BIC: two-parameter original HMM: BIC = 3562.8; four-parameter ntHMM: BIC = 4276.9; relative likelihood (BIC weight) of the four-parameter ioHMM <10^–155). In the light of the model comparison results, we decided to fit the simpler two-parameter HMM model.

## Analyzing model dynamics

In order to understand how exploration and exploitation changed across males and females, we analyzed the HMMs. The term 'dynamics' means the rules or laws that govern how a system evolves over time. Here, the system of interest was decision making behavior, specifically at the level of the hidden explore and exploit goals. In fitting our HMMs, we were fitting a set of equations that describe the dynamics of these goals: the probability of transitions between exploration and exploitation and vice versa. Of course, having a set of fitted equations is a far cry from understanding them. To develop an intuition for how sex altered the dynamics of exploration, we therefore turned to analytical tools that allowed us to directly characterize the energetic landscape of behavior (*Figure 1H*).

In statistical mechanics, processes within a system (like a decision-maker at some moment in time) occupy states (like exploration or exploitation). States have energy associated with them, related to the long-time scale probability of observing a process in those states. A low-energy state is one that is very stable and deep, much like a valley between two high peaks. Low-energy states will be over-represented in the system. A high energy state, like a ledge on the side of a mountain, is considerably less stable. High-energy states will be under-represented in the long-term behavior of the system. The probability of observing a process in a given state i will is related to the energy of that state ($E_i$) via the Boltzman distribution:

$$p_i = \frac{1}{Z} e^{\frac{-E_i}{k_B T}}$$

where Z is the partition function of the system, $k_B$ is the Boltzman constant, and T is the temperature of the system. If we focus on the ratio between two state probabilities, the partition functions cancel out and the relative occupancy of the two states is now a simple function of the difference in energy between them:

$$\frac{p_i}{p_j} = e^{\frac{-\left(E_i - E_j\right)}{k_B t}}$$

Rearranging, we can now express the difference in energy between two states as a function of the difference in the long-term probability of those states being occupied:

$$ln\left(\frac{p_i}{p_j}\right) k_B T = E_j - E_i$$

Meaning that the difference in the energetic depth of the states is proportional to the natural log of the probability of each state, up to some multiplicative factor $k_B$T. To calculate the probability of

exploration and exploitation ($p_i$ and $p_j$), we calculated the stationary distribution of the fitted HMMs. The stationary distribution is the equilibrium probability distribution over states. This means that this distribution is the relative frequency of each state that we would observe if the model's dynamics were run for an infinite period of time. Each entry of the model's transition matrix reflects the probability that the mice would move from one state (e.g. exploring) to another (e.g. exploiting one of the options) at each moment in time. Because the parameters for all the exploitation states were tied, each transition matrix effectively had two states—an explore state and a generic exploit that described the dynamics of all exploit states. Each of the k animals had its own transition matrix ($A_k$), which describes how the entire system—an entire probability distribution over states—would evolve from time point to time point. We observe how the dynamics evolve any probability distribution over states ($\pi$) by applying the dynamics to this distribution:

$$\pi_{t+1} = \pi_t A_k$$

Over many time steps, ergodic systems will reach a point where the state distributions are unchanged by continued application of the transition matrix as the distribution of states reaches its equilibrium. That is, in stationary systems, there exists a stationary distribution, $\pi^*$, such that:

$$\pi = \pi A_k$$

If it exists, this distribution is a (normalized) left eigenvector of the transition matrix $A_k$ with an eigenvalue of 1, so we solved for this eigenvector to determine the stationary distribution of each $A_k$. We then took an average of these stationary distributions within each sex, plugged them back into the Boltzman equations to calculate the relative energy (depth) of exploration and exploitation as illustrated in *Figure 1H*.

In order to understand the dynamics of our coarse-grained system, we need to not only understand the depth of the two states, but also the height of the energetic barrier between them: the activation energy required to transition from exploration to exploitation and back again. Here, we build on an approach from chemical kinetics that relates the rate of transition between different conformational states to the energy required to affect these transitions. The Arrhenius equation relates the rate of transitions away from a state (k) to the activation energy required to escape that state ($E_a$):

$$k = A e^{\frac{E_a}{k_B T}}$$

where A is a constant pre-exponential factor related to the readiness of reactants to undergo the transformation. We will set this to one for convenience. Again, $k_B T$ is the product of temperature and the Boltzman constant. Note the similarities between this equation and the Boltzman distribution illustrated earlier. Rearranging this equation to solve for activation energy yields:

$$E_a = -ln\left(\frac{k}{A}\right) k_B T$$

Thus, much like the relative depth of each state, activation energy is also proportional to some measurable function of behavior, up to some multiplicative factor $k_B T$. Note that our approach has only identified the energy of three discrete states (an explore state, an exploit state, and the peak of the barrier between them). These are illustrated by tracing a continuous potential through these three points only to provide a physical intuition for these effects.

## Reinforcement learning models

We fitted seven reinforcement learning (RL) models that could potentially characterize animals' choice behaviors, with details of each RL model as below. To identify the model that best captured the computations used by the animals, we compared model fits across six reinforcement learning models with different combinations of latent parameters. AIC weights were calculated from AIC values of each model for each sex and compared across models to determine the best model with the highest relative likelihood.

The first model assumes that animals choose between two arms randomly with some overall bias for one side over the other. This choice bias for choosing left side over right side is captured with a parameter *b*. The probability of choosing left side on trial *t* is:

$$p_t^L = b \qquad \text{[1] "random"}$$

The second model is a noisy win-stay lose-shift (WSLS) model that adapts choices with regards to outcomes. This model assumes a win-stay lose-shift policy that is to repeat a rewarded choice and to switch to the other choice if not rewarded. Furthermore, this model includes a parameter $\epsilon$ that captures the level of randomness, allowing a stochastic application of the win-stay lose-shift policy. The probability of choosing arm $k$ on trial $t$ is:

$$p_t^k = \begin{cases} 1 - \frac{\epsilon}{2}, if\left(c_{t-1} = k \wedge r_{t-1} = 1 \vee c_{t-1} \neq k \wedge r_{t-1} = 0\right) \\ \frac{\epsilon}{2}, if\left(c_{t-1} \neq k \wedge r_{t-1} = 1 \vee c_{t-1} = k \wedge r_{t-1} = 0\right) \end{cases} \text{[2] "noisy WSLS"}$$

$c_t$ indicates the choice on trial $t$ and $r_t$ is a binary variable that indicates whether or not trial $t$ was rewarded.

The third model is a basic delta-rule reinforcement learning (RL) model. This two-parameter model assumes that animals learn by consistently updating Q values, which are values defined for options (left and right side). These Q values, in turn, dictate what choice to make next. For example, in a multi-armed bandit task, $Q_t^k$ is the value estimation of how good arm $k$ at trial $t$, and is updated based on the reward outcome of each trial:

$$Q_{t+1}^k = Q_t^k + \alpha\left(r_t - Q_t^k\right) \qquad \text{[3] "RL"}$$

In each trial, $r_t - Q_t^k$ captures the reward prediction error (RPE), which is the difference between expected outcome and the actual outcome. The parameter $a$ is the learning rate, which determines the rate of updating RPE. With Q values defined for each arm, choice selection on each trial was performed based on a Softmax probability distribution:

$$p\left(a_{t+1} = k\right) = \frac{e^{\beta Q_t^k}}{\sum_j e^{\beta Q_t^j}}$$

, where inverse temperature $\beta$ determines the level of random decision noise.

The fourth model incorporates a lapse rate parameter ($\varepsilon$), which reduces the influence of value-independent choices on the estimation of the remaining parameters, capturing any noises outside the softmax value function.

$$p\left(a_{t+1} = k\right) = \varepsilon + \left(1 - 2\varepsilon\right)\frac{e^{\beta Q_t^k}}{\sum_j e^{\beta Q_t^j}} \qquad \text{[4] "RL$\varepsilon$"}$$

The fifth model incorporates a choice updating rules in addition to the value updating rule in model 3. The model assumes that choice kernel, which captures the outcome-independent tendency to repeat a previous choice, also influences decision making. The choice kernel updating rule is similar to the value-updating rule:

$$CK_{t+1}^k = CK_t^k + \alpha_c\left(a_t^k - CK_t^k\right) \qquad \text{[5] "RLCK"}$$

, where $a_t^k$ is a binary variable that indicates whether or not arm $k$ was chosen on trial $t$ and $a_t$ is choice kernel updating rate, characterizing choice persistence. The value and choice kernel term were combined to compute the probability of choosing arm $k$ on trial $t$:

$$p_t^k = \frac{e^{\left(\beta Q_t^k + \beta_c CK_t^k\right)}}{\sum_j e^{\left(\beta Q_t^j + \beta_c CK_t^j\right)}}$$

, where $\beta_c$ is the inverse temperature associated with the choice kernel, capturing the stickiness of choice.

The sixth model is the same as the fourth model, except that this model includes another parameter $\gamma$ that modulates learning rate when the choice is not rewarded. This model assumes asymmetrical learning that the learning rate is different for rewarded and unrewarded trials.

$$Q_{t+1}^k = \begin{cases} Q_t^k + \alpha\left(r_t - Q_t^k\right), r_t = 1 \\ Q_t^k + \gamma \times \alpha\left(r_t - Q_t^k\right), r_t = 0 \end{cases}$$ [6] "RLCKγ"

The seventh model is also similar to model 4, except that this model includes another parameter $\eta$ that tunes the balance between the value updating rule and the choice kernel updating rule. This model assumes that animals could be using two policies (value and choice kernel) to different extent, that is some animals could depend their choices more heavily on values and some animals could be more dependent on choice preference. In this model, the probability of choosing arm $k$ on trial $t$:

$$p_t^k = \frac{e^{\left(\eta\beta Q_t^k + (1-\eta)\beta_c CK_t^k\right)}}{\sum_j e^{\left(\eta\beta Q_t^j + (1-\eta)\beta_c CK_t^j\right)}}$$ [7] "RLCK $\eta$ "

## Conditional mutual information

We quantified the extent to which choice history was informative about current choices as the conditional mutual information between the current choice ($C_t$) and the last choice ($C_{t-1}$), conditioned on the reward outcome of the last trial (R):

$$I\left(C_t; C_{t-1}|R\right) = \sum_{r \in R} \sum_{c_{t-1} \in C} \sum_{C_t \in C} p_{C_t, C_{t-1}, R}\left(c_t, c_{t-1}, r\right) \log \frac{p_R(r) p_{C_t, C_{t-1}, R}\left(c_t, c_{t-1}, r\right)}{p_{C_t, R}\left(c_t, r\right) p_{C_{t-1}, R}\left(c_{t-1}, r\right)}$$

where the set of choice options (C) represented the two options (left/right).

## Acknowledgements

This work was supported by NIMH R01 MH123661, NIMH P50 MH119569, and NIMH T32 training grant MH115886, startup funds from the University of Minnesota (NMG), a Young Investigator Grant from the Brain and Behavior Foundation (RBE), an Unfettered Research Grant from the Mistletoe Foundation (RBE), and Fonds de Recherche du Québec Santé, Chercheur-Boursier Junior 1, #284,309 (RBE). Thank you to Briana E Mork and Lisa S Curry-Pochy for help improve this manuscript.

## Additional information

### Funding

| Funder | Grant reference number | Author |
| --- | --- | --- |
| National Institutes of Health | R01MH123661 | Nicola M Grissom |
| National Institutes of Health | P50MH119569 | Nicola M Grissom |
| Brain and Behavior Research Foundation | | R Becket Ebitz |
| Mistletoe Foundation | | R Becket Ebitz |
| Fonds de Recherche du Québec - Santé | | R Becket Ebitz |
| University of Minnesota | University of Minnesota start-up funds | Nicola M Grissom |

The funders had no role in study design, data collection and interpretation, or the decision to submit the work for publication.

### Author contributions

Cathy S Chen, Conceptualization, Formal analysis, Investigation, Methodology, Validation, Visualization, Writing - original draft, Writing – review and editing; Evan Knep, Investigation, Writing - original

draft, Writing – review and editing; Autumn Han, Investigation; R Becket Ebitz, Conceptualization, Funding acquisition, Methodology, Supervision, Writing - original draft, Writing – review and editing; Nicola M Grissom, Conceptualization, Funding acquisition, Methodology, Supervision, Writing – review and editing

### Author ORCIDs
Cathy S Chen ![ID] http://orcid.org/0000-0003-2506-8522
Nicola M Grissom ![ID] http://orcid.org/0000-0003-3630-8130

### Ethics
This study was performed in strict accordance with the recommendations in the Guide for the Care and Use of Laboratory Animals of the National Institutes of Health. All of the animals were handled according to approved institutional animal care and use committee (IACUC) protocols (#1912-37717A) of the University of Minnesota.

### Decision letter and Author response
Decision letter https://doi.org/10.7554/eLife.69748.sa1
Author response https://doi.org/10.7554/eLife.69748.sa2

## Additional files

### Supplementary files
• Supplementary file 1. Four-way Repeated Measures ANOVA showing main effects and interaction effects (pairwise and 3-way) of sex, outcome, latent state, and subject identity.

• Supplementary file 2. Tukey's multiple comparison test of model agreement across RL models showing how well each model predicts animals' actual choices.

• Transparent reporting form

### Data availability
All behavioral data have been deposited in generic database (Dyrad) with accession link https://doi.org/10.5061/dryad.z612jm6c0.

The following dataset was generated:

| Author(s) | Year | Dataset title | Dataset URL | Database and Identifier |
|---|---|---|---|---|
| Chen CS | 2021 | Sex differences in learning from exploration | https://doi.org/10.5061/dryad.z612jm6c0 | Dryad Digital Repository, 10.5061/dryad.z612jm6c0 |

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
