## [Editor Report]

Following inclusion of new modeling and data presentation, authors have more clearly demonstrated that equivalent performance is seen across males and females in terms of reward rate, yet achieved via different successful strategies. This is an important contribution to the growing literature on sex differences in reinforcement learning.

---

## [Decision Letter]

**Decision letter after peer review:**

Thank you for submitting your article "Sex differences in learning from exploration" for consideration by *eLife*. Your article has been reviewed by 3 peer reviewers, including Alicia Izquierdo as Reviewing Editor and Reviewer #1, and the evaluation has been overseen by Kate Wassum as the Senior Editor.

Essential revisions:

Chen et al., trained male and female animals on an explore/exploit (2-armed bandit) task. Despite similar levels of accuracy in these animals, authors report higher levels of exploration in male than in female mice. The patterns of exploration were analyzed in fine-grained detail: males are less likely to stop exploring once exploring is initiated, whereas females stop exploring once they learn. Authors find that both learning rate (α) and noise parameter (β) increase in exploration trials in a hidden Markov model (HMM). When reinforcement learning (RL) models were fitted to animal data, they report females had a higher learning rate and over days of testing, suggesting higher meta-learning in females. They also report that of the RL models they fit, the model incorporating a choice kernel updating rule was found to fit both male and female learning. The results suggest one should pay greater attention to the influence of sex in learning and exploration. Another important takeaway from this study is that similar levels of accuracy do not imply similar strategies. There are 2 categories of essential revisions suggested by Reviewers:

1) There was a general concern that reframing of conclusions may be warranted due to the major results possibly reflecting learning more than exploration. Female rats may learn the task better than male rats. For more clarity on this issue, reviewers request that authors present more primary behavioral data (p(reward,obtained) vs time (days), reaction times over time, etc.) to justify their conclusions. It was also unclear how reaction times were calculated and how "steady state" was operationalized.

2) Reviewers also asked for better justification and details for both the hidden Markov model and reinforcement learning parameters. If for example, male rats simply learn the task more poorly and behave more randomly, this would manifest as more exploration in the HMM model. Additional analyses are needed to strengthen authors' claims using the HMM model- the effect of obtained reward on state transitions, and biased exploitations should be further explored as there are presently a number of unjustified assumptions.

Please address these essential concerns (which are detailed in the reviews below), as well as the reviewers other comments.

*Reviewer #1 (Recommendations for the authors):*

Chen et al., trained male and female animals on an explore/exploit (2-armed bandit) task. Despite similar levels of accuracy in these animals, authors report higher levels of exploration in males than in females. The patterns of exploration were analyzed in fine-grained detail: males are less likely to stop exploring once exploring is initiated, whereas female mice stop exploring once they learn. Authors find that both learning rate (α) and noise parameter (β) increase in exploration trials in a hidden Markov model (HMM). When reinforcement learning (RL) models were fitted to animal data, they report females had a higher learning rate and over days of testing, suggesting higher meta-learning in females. They also report that of the RL models they fit, the model incorporating a choice kernel updating rule was found to fit both male and female learning. The results do suggest one should pay greater attention to the influence of sex in learning and exploration. Another important takeaway from this study is that similar levels of accuracy do not imply similar strategies. I have suggestions for clarity in data presentation and interpretation.

One of the first sections in the Results section dives straight away into the HMM, but in my opinion, authors do not present enough of the primary behavioral data- perhaps I missed this, but can we see p(reward, obtained) over sessions for males and females (more information than Figure 1B)? And the reaction times in Figure 1C, are these reaction times to make a left/right response or reaction times to collect rewards? Can authors show both, as a function over time?

What is the cited rationale for the different RL models and their parameters? If the RLCK is the best fit for both males and females, does this lend support to the idea that though overall learning many not differ between males and females, the strategies are not well captured by RL? Please clarify.

Authors should clarify the difference between learning and "steady state." How was this operationally defined and measured? This was a bit lost in the data presentation.

The lines 430-432 about rodent behavioral tasks are unclear to me: "However, the vast majority of these tasks were not designed with computational models in mind, and as a result, we are unable to assess whether similar latent cognitive variables are influencing behavior in humans and rodents." There are several groups that use touchscreen-response methods paired with computational modeling. Do the authors mean they do not have access to similar databases to compare these latent variables across species? Authors may want to clarify how these experiments uniquely identify latent cognitive variables not previously explored with similar methods.

*Reviewer #2 (Recommendations for the authors):*

1. How is reaction time computed here? Do you remove outliers (extremely long RTs)? Is there a way to separate exploring from guessing in RT (given that behaviorally they are confounded)?

2. State transitions are not value dependent in the HMM model. Another value independent way of "exploration" is by having a lapse rate in the RL model. I am curious about whether there is a lapse rate difference across sex (and possibly no differences in the temperature term).

3. There is the inset panel (ROC curve) in all density figures except Figure 3D.

4. I like your dynamic landscape illustration of the fitted HMM (Figure 1G).

*Reviewer #3 (Recommendations for the authors):*

(1) A great proportion of reported results and analysis rely on the extracted latent states from the proposed HMM. While HMM is proposed to provide a model free analysis of behavior, certain choices regarding the HMM model need further justification:

(1.1) Most importantly, authors assume that only most recent state determines the state in the next trial. However, I argue that most recent obtained reward is another determinant of the state in the next trial and should be added to the model. This way instead of using a naïve HMM, and then exploring learning in explore/exploit trials, authors can compare HMM parameters.

(1.2) Proposed HMM model also assumes that exploit states are uniform across the options. Do authors have any evidence supporting this assumption? Side biases are commonly observed in animals and humans. Extracted RL parameters also confirm this. Please comment.

(1.3) Moreover, the model assumes that the mice had to pass through exploration in order to start exploiting a new option. Do authors have any evidence supporting this assumption? What will happen to the results if this assumption is lifted? Please comment.

(2) Page 10, line 252: Please provide a more quantitative comparison of models' choice behavior (and not just RLCK) and the animals' behavior for all sessions. Also, there are no tick-marks on the y-axis.

(3) How much overlap exists between the extracted latent dynamics from HMM and that of previously proposed models mentioned in the methods (Daw et al., 2006; Jepma and Nieuwenhuis, 2011; Pearson et al., 2009)? It would be helpful to show the extent that results from these different methods deviate/overlap with each other.

(4) Do authors see any differences between amount of exploration/exploitation at the beginning vs at the end of a session? How about across days? The fact that meta learning is observed, suggests that even in a single session, changes in the strategy of animals might be expected.

---

## [Author Response]

1) There was a general concern that reframing of conclusions may be warranted due to the major results possibly reflecting learning more than exploration. Female rats may learn the task better than male rats. For more clarity on this issue, reviewers request that authors present more primary behavioral data (p(reward,obtained) vs time (days), reaction times over time, etc.) to justify their conclusions.

The reviewers were concerned that the apparent sex differences in exploration were actually the result of sex differences in how well the animals learn the task, which we have addressed in two ways. First, we have improved the clarity of the manuscript demonstrating that there were no differences in how “well” the male and female mice performed the task in terms of reward acquisition. The revised manuscript indeed shows that (1) male and female animals had equivalent levels of performance in terms of rewards earned, and (2) males and females both reached equivalent performance levels on the bandit task before comparisons of their strategies began. As we detail below, we have revised the manuscript to make this point clearer, including adding the requested primary behavioral data (Supplemental Figure 1).

Second, this comment also highlights a deeper point--that learning (technically defined as a process of updating expectations) and exploration (defined as a process of environment sampling) are often conflated and understanding the precise relationship between the two is essential. Further, these measures are often conflated with reward performance (that is, “better reward performance means better learning”), but reward performance is at its core a dependent measure that can reflect different combinations of contributions from the cognitive processes of learning/updating expectations and exploration/sampling the environment. We believe that the insights we provide into the relationship between learning and exploration is a strength of the revised manuscript, not a misinterpretation. We show that trial-by-trial measures of learning are intrinsically linked to exploration, though these links differ between males and females. The reviewer’s comments lead to the interesting question of whether learning differences beget exploration differences, whether the reverse is true, or whether these latent cognitive variables are dissociable from each other. Unfortunately, we cannot yet say whether differences in learning drive differences in exploration or vice versa, but we highlight this issue for future research in the discussion.

It was also unclear how reaction times were calculated and how "steady state" was operationalized.

We apologize for omitting an operational definition of reaction times. The revised manuscript explains that response times were calculated as the time elapsed between choice display onset and the time when the nose poke response was completed, as measured by the touch screen.

Regarding the term “steady state”, we believe that this comment is related to Reviewer #1’s major comment: “Authors should clarify the difference between learning and "steady state." How was this operationally defined and measured? This was a bit lost in the data presentation.” We apologize that we could not find a place in the manuscript where we used the term steady-state ourselves, so we present a few ways to hopefully address this point. First, we considered the possibility that the reviewer was referring to asymptotic steady-state behavior (i.e. whether the animals had learned the task before data collection began). If this reading of the comment is correct, then we hope that our other comments and new analyses regarding asymptotic performance levels will address this point. We also considered the alternative possibility that the reviewer may have in mind a different version of a bandit task (e.g.: reversal task, in which steady-state periods in reward contingencies occur prior to reversals). To address this possible concern, we have clarified that there is no steady state component in this task: the reward probabilities of each choice continuously changed over time, so the animals were encouraged to continuously learn about the most rewarding choice(s). These changes are detailed in our response to this comment below.

2) Reviewers also asked for better justification and details for both the hidden Markov model and reinforcement learning parameters. If for example, male rats simply learn the task more poorly and behave more randomly, this would manifest as more exploration in the HMM model.

The reviewers were concerned that the sex difference in HMM-inferred exploration could merely be due to poor learning performance in males. In our brief response to the previous comment and detailed comments below, we explain that male mice did not perform the task more poorly than females and several new analyses show that there were no sex differences in how well the task was performed.

It is important to clarify that there are a great number of equivalently good strategies in this task, some even involving random choosing (Daw et al., 2006; Sutton and Barto, 1998; Wilson et al., 2021). This happens because this task is so dynamic and the best option so uncertain. From this vantage point, an animal behaving randomly, if possible, could be a valid approach However, this subtle point was a bit lost in the original manuscript, so we have made several revisions to try to bring it to the forefront. We have added new figures and supplemental figures, as well as changed in text in results and discussion. We hope these revisions clarify that the critical observation we make here is not that either sex performed more poorly--instead, males and females received similar amounts of reward through fundamentally different strategies.

We recognize that the confusion might have come from the result of the reinforcement learning (RL) model, where females had higher learning rates (ɑ) than males, indicating differences in the rate of updating value expectations. In the revised manuscript, we clarified that the learning rate parameter in the RL model is not equivalent to the dependent measure of reward performance, which is better indicated by the number of rewards obtained. We have added a new figure panel to Figure 3 to illustrate two points. First, that higher learning rate is not equivalent to more reward earned. In fact, it is possible to learn so rapidly (that is, make large, sudden adjustments to expectations of value) that performance is actually compromised in a stochastically rewarding environment! Second, this new panel illustrates that different combinations of the RL model parameters learning rate (ɑ) and decision noise (β) could result in similar performance. We have also rewritten our discussion of the significance of the sex difference in learning rate parameter results to clarify our interpretation and ensure that our claims are more precise.

Additional analyses are needed to strengthen authors' claims using the HMM model- the effect of obtained reward on state transitions, and biased exploitations should be further explored as there are presently a number of unjustified assumptions.

The reviews requested detailed justification for the specific HMM we used in the original manuscript. We agree that the HMM was an essential part of the manuscript and the choices we made about its structure deserved a more detailed justification. In the revised manuscript, we include a formal comparison of the simplified model we originally fit with two, more complex versions of the HMM model: an input-output HMM (ioHMM) and an unrestricted HMM, fit without any parameter tying (ntHMM). The input-output HMM model takes into account the effect of reward on choice dynamics. And the unrestricted HMM allows for bias exploitations. However, model comparison revealed that there was no benefit to these more complex models, either in terms of their ability to fit the data, or in terms of their ability to explain variance in choice behavior. The original HMM was the most parsimonious model and as a result it is the best able to explain choice dynamics. These results are included in the revised manuscript, along with additional text that explains why the HMM was structured as it was (details in the comments below).

Please address these essential concerns (which are detailed in the reviews below), as well as the reviewers other comments.Reviewer #1 (Recommendations for the authors):Chen et al., trained male and female animals on an explore/exploit (2-armed bandit) task. Despite similar levels of accuracy in these animals, authors report higher levels of exploration in males than in females. The patterns of exploration were analyzed in fine-grained detail: males are less likely to stop exploring once exploring is initiated, whereas female mice stop exploring once they learn. Authors find that both learning rate (α) and noise parameter (β) increase in exploration trials in a hidden Markov model (HMM). When reinforcement learning (RL) models were fitted to animal data, they report females had a higher learning rate and over days of testing, suggesting higher meta-learning in females. They also report that of the RL models they fit, the model incorporating a choice kernel updating rule was found to fit both male and female learning. The results do suggest one should pay greater attention to the influence of sex in learning and exploration. Another important takeaway from this study is that similar levels of accuracy do not imply similar strategies. I have suggestions for clarity in data presentation and interpretation.One of the first sections in the Results section dives straight away into the HMM, but in my opinion, authors do not present enough of the primary behavioral data- perhaps I missed this, but can we see p(reward, obtained) over sessions for males and females (more information than Figure 1B)? And the reaction times in Figure 1C, are these reaction times to make a left/right response or reaction times to collect rewards? Can authors show both, as a function over time?

Thank you for this important feedback. In the revised manuscript, we have included more primary behavioral data in Supplemental Figure 1, including reward acquisition over sessions, response time over sessions, and reward retrieval time over sessions. The addition of these new figures demonstrates two points more clearly: (1) males and females had identical reward acquisition performance and neither sex was performing poorer than the other; (2) multiple behavioral metrics (reward acquisition, response time, reward retrieval time) converge to show that animals had reached asymptotic performance. We have also clarified those points in the manuscript: that animals have learned how to perform the task before data collection began, whereas we are focused on understanding how animals continued to make flexible decisions in a well-learned, but non-stationary environment.

Additionally, we have added more text to explain how response time and reward retrieval time was calculated. Response times were calculated as the time elapsed between choice display onset and the time when the nose poke response was completed, as measured by the touch screen. Reward retrieval time was calculated as the time elapsed between the nose-poke response for choice and magazine entry for reward retrieval. We apologize for omitting this definition in the original manuscript.

New text from the manuscript (page 4, line 93-113):

“It is worth noting that unlike other versions of bandit tasks such as the reversal learning task, in the restless bandit task, animals were encouraged to continuously learn about the most rewarding choice(s). […] Since both sexes have learned to perform the task prior to data collection, variabilities in task performance are results of how animals learned and adapted their choices in response to the changing reward contingencies.”

(page 6, line 133-140):

“Therefore, we examined the response time, which was calculated as time elapsed between choice display onset and nose poke response as recorded by the touchscreen chamber, in both males and females. If males and females had adopted different strategies here, then we might expect response time to systematically differ between males and females, despite the similarities in learning performance. Indeed, females responded significantly faster than did males (Figure 1C, main effect of sex, t(30) = 3.52, p = 0.0014), suggesting that decision making computations may differ across sexes and, if so, that the strategies that tended to be used by females resulted in faster choice response time than those used by males.”

What is the cited rationale for the different RL models and their parameters? If the RLCK is the best fit for both males and females, does this lend support to the idea that though overall learning many not differ between males and females, the strategies are not well captured by RL? Please clarify.

This comment raises an interesting point and an important concern. First, to address the concern, we have added more clarifying text to explain the rationale of fitting different reinforcement learning (RL) models and added relevant citations. This is detailed below. Second, to the interesting point: we too had considered that strategy differences between male and female animals might mean that they have different best-fitting RL models. However, the fact that both were best fit by the same RLCK model does not mean that they did not have different strategies or that RL modeling cannot capture those strategies. It may, however, suggest that strategic differences between the sexes are more a matter of degree (i.e. of differences in the specific pattern of model parameters), rather a matter of categorically distinct computations. This interpretation also makes the most sense in light of the biology of sex differences, which produce few (if any) truly categorically distinct changes in neural function, but rather serve to bias neural systems across sexes in multiple complex ways.

In terms of why we chose to look at the specific RL models that we did, we have revised the manuscript to explain (1) what different flavors of RL model are commonly used in the decision making literature, and (2) why we chose the specific models we did. We have added 1 additional model, per another review’s request, but the relevant text reads as follows (page 12, line 305-340):

“The HMM suggested that males and females had different levels of exploration, but it did not provide insight into the latent, cognitive processes behind these differences. […] This interpretation also makes the most sense in light of the biology of sex differences, which produce few (if any) truly categorically distinct changes in neural function, but rather serve to bias neural systems across sexes in multiple complex ways.”

Authors should clarify the difference between learning and "steady state." How was this operationally defined and measured? This was a bit lost in the data presentation.

We apologize that we could not find a place in the manuscript where we used the term steady-state ourselves, so we present a few ways to hopefully address this point.

First, we considered the possibility that the reviewer was referring to asymptotic steady-state behavior (i.e. whether the animals had learned the task before data collection began). In the revised manuscript, we first clarified that there are two types of learning that could occur in this task: (1) global learning about the statistics of the reward environment, e.g.: how frequently do reward probabilities change; (2) learning about the most valuable option at each time point. In the revised manuscript, we clarified and included a supplemental figure to demonstrate that the global learning of the task structure has reached asymptotic levels in both sexes (no changes in reward acquisition, response time, reward retrieval time across sessions) and we are examining how animals explored and learned about which option currently provided the best reward probability.

Second, we considered the alternative possibility that the reviewer may have in mind a different version of a bandit task (e.g.: reversal task, in which there exists steady periods in reward contingencies prior to reversals). In any case, it was important to clarify that there was no steady state component in this task: the reward probabilities of each choice continuously changed over time, so the animals were encouraged to continuously learn about the most rewarding choice(s). We have clarified our descriptions of the task to better explain the dynamic structure of the task and to highlight the differences between this task and classic reversing reward schedules.

The lines 430-432 about rodent behavioral tasks are unclear to me: "However, the vast majority of these tasks were not designed with computational models in mind, and as a result, we are unable to assess whether similar latent cognitive variables are influencing behavior in humans and rodents." There are several groups that use touchscreen-response methods paired with computational modeling. Do the authors mean they do not have access to similar databases to compare these latent variables across species? Authors may want to clarify how these experiments uniquely identify latent cognitive variables not previously explored with similar methods.

We apologize for the poor choice of phrasing here and hope the revised manuscript clarifies our point: that we are excited to be a part of an emerging and translationally important trend in rodent computational neuroscience, not that we are the only group working in this space. We meant only to distinguish this trend from the historical use of species-specific tasks in translational research. New text from the manuscript (page 19, line 542-550):

“Rodent operant testing is frequently used to assess cognitive functions. This is critical for translational work in animals needed to link pharmacology, genetics, and other potential biological contributors to behavior (Grissom and Reyes, 2018; Heath et al., 2016). However, many classic rodent cognitive tasks are species-specific: they were not designed to assess the same cognitive processes across species and this limits their translational ability. Currently, there is an emerging exciting trend among rodent researchers with adopting tasks with translational potentials in rodents, such as reversal learning and various versions of bandit task (Izquierdo et al., 2019; Groman et al., 2016; Bari et al., 2019; Grossman et al., 2020), where we can assess the same cognitive processes across species and across animal models of diseases.”

Reviewer #2 (Recommendations for the authors):1. How is reaction time computed here? Do you remove outliers (extremely long RTs)? Is there a way to separate exploring from guessing in RT (given that behaviorally they are confounded)?

We apologize for omitting the definition of response time. The revised manuscript explains that response times were calculated as the time elapsed between choice display onset and the time when the nose poke response was completed, as measured by the touch screen. Z scores of response times for each animal were calculated and response times with z scores above or below 3 standard deviations from the mean were removed as outliers. As we responded earlier in comment 1B, we are not able to differentiate between exploring and guessing with our models in this task.

2. State transitions are not value dependent in the HMM model. Another value independent way of "exploration" is by having a lapse rate in the RL model. I am curious about whether there is a lapse rate difference across sex (and possibly no differences in the temperature term).

We take the reviewer’s point that there may be other noise/bias outside the softmax value function. In the revised manuscript, we have included a new RL model that incorporates the lapse rate. However, the model comparison (AIC) showed that the new RL lapse rate model did not improve the model fit (Figure 3B). In addition, we also examine the model agreement of this new RL lapse rate model to determine whether including a lapse rate parameter improves the model’s ability to predict the animals’ choice. However, multiple comparisons suggested that including the lapse rate parameter did not improve the model’s ability to predict animals’ choice (Figure 3C, Supplemental Table 2, mean difference < 0.0004, p > 0.99). Based on this result, we believe the RLCK model we used in the original manuscript, which captures both value and value independent choice bias, is still the simplest model that best captured and predicted animals’ actual choice.

New text from the manuscript (page 12, line 323-324):

“(4) a three-parameter “RLε” model with a consistent learning rate, and inverse temperature that captures value-based decision noise, and a value-independent noise”

page 13, line 341-356:

“To quantify how well each RL model was at predicting animals’ choices, we measured the model agreement for each model, which was calculated as the probability of choice correctly predicted by the optimized model parameters for each model (Figure 3C). Then we conducted a multiple comparison across model agreement of RL models (test statistics reported in Supplemental Table 2). The results suggested that the RL models with parameter(s) that account for choice bias (RLCK, RLCKγ, RLCKη) were significantly better at predicting animals’ actual choices than the models that do not account for choice bias and non-RL models (random, noisy WSLS, RL, RLε). There was no significant difference in model agreement between RLCK, RLCKγ, and RLCKη. Based on the result of model comparison (AIC) and model agreement, we decided that the four-parameter RLCK model is the simplest, best-fit model that best predicted animal’ actual choices. Finally, to visualize how well the RLCK model was at predicting choices of animals with different learning performance, we plotted the simulated choices and actual choices against the matching law (Poling et al., 2011), which dictates that the probability of choice is proportional to the probability of reward. The figure showed that this four-parameter model was able to characterize animals’ choice behaviors regardless of the value-based learning performance (Supplemental Figure 3).”

page 37, line 927-932:

“The fourth model incorporates a lapse rate parameter (ε), which reduces the influence of value-independent choices on the estimation of the remaining parameters, capturing any noises outside the softmax value function.

p(at+1=k)=ε+(1−2ε) eβQtk∑jeβQtj[4] “RLε”

3. There is the inset panel (ROC curve) in all density figures except Figure 3D.

The inset ROC curve graph has been added to what’s now Figure 3E. Thank you for pointing it out.

4. I like your dynamic landscape illustration of the fitted HMM (Figure 1G).

Thank you!

Reviewer #3 (Recommendations for the authors):Major concerns:1) A great proportion of reported results and analysis rely on the extracted latent states from the proposed HMM. While HMM is proposed to provide a model free analysis of behavior, certain choices regarding the HMM model need further justification:

We have added additional information on our HMM model design. Briefly, each of these choices simplified the model. In part, we wanted to be mindful of the tradeoff between fitting the data as well as possible and making a simpler, more interpretable model. We erred on the side of the latter here because we had small amounts of data, relatively speaking (which makes fitting more parameters more challenging), and were interested in interpreting specific parameters. In the light of the model comparison results, we decided to stick with the simpler 2-parameter HMM model in the original manuscript. Unfortunately, we do not have enough data to accurately estimate more parameters than we would like to. We believe that in the future experiments oriented towards capturing a large amount of data from our animals, we would be able to fit more sophisticated HMM models that allow for more flexibility to estimate input output relationships, and specific side-based exploration, etc. Please see the detailed responses below.

(1.1) Most importantly, authors assume that only most recent state determines the state in the next trial. However, I argue that most recent obtained reward is another determinant of the state in the next trial and should be added to the model. This way instead of using a naïve HMM, and then exploring learning in explore/exploit trials, authors can compare HMM parameters.

We have extended the model in the ways suggested by the reviewer, but these changes did not improve model fit. We nevertheless share the reviewer’s intuition and suspect that we might be able to fit more complex models if we had more data. However, given that we had ~2k trials per animal, simpler models had an advantage here. The relevant text reads:

New text from the manuscript (page 8, line 203-211):

“Since various factors could influence state of the next trial, we considered a simple 2 parameter HMM that models only two states (exploration and exploitation), a 4-parameter input-output HMM (ioHMM) that allows reward outcome to influence the probability of transitioning between states, and a 4-parameter unrestricted HMM with no promoter tying (ntHMM) that allows biased exploitation (see methods). The model comparisons have shown that the 2 parameter HMM was the simplest, most interpretable, and best fit model (AIC: 2 parameter HMM, AIC = 2976.1; ioHMM, AIC = 3117; ntHMM, AIC = 3101.5, see more statistics reported in Methods). Therefore, we selected the simple 2-parameter HMM to infer the likelihood that each choice was part of the exploratory regime, or the exploitative one (see Methods).”

page 33, line 792-807:

“To account for the effect of reward on choice dynamics, we extended the 2-parameter HMM model to an input-output HMM model (4-parameter ioHMM), whose framework allows inputs, such as reward outcomes, to influence the probability of transitioning between states (Bengio and Frasconi ; Ebitz et al., 2019). […] While reward outcomes could affect the probability of state transitions, the model comparison suggested that the extra input layer did not explain more variance in choice dynamics, and therefore, we would favor the original, simpler 2-parameter HMM model.”

(1.2) Proposed HMM model also assumes that exploit states are uniform across the options. Do authors have any evidence supporting this assumption? Side biases are commonly observed in animals and humans. Extracted RL parameters also confirm this. Please comment.

We have included a new HMM that allows for bias exploitation as suggested by the reviewer, but this change did not improve model fit. We absolutely agree with the reviewers that exploitation can be biased, as mice do display side bias to some extent. One issue we face with an unrestricted HMM model with no parameter tying is that the unfavored choice would have fewer trials for parameter estimation, especially in animals with strong side bias. Tying the parameters in the HMM model is a common practice for more accurate parameter estimation, but we recognize the limitations. Here, in our original model, tying the parameters allowed us to more accurately estimate the parameters of the transition matrix using information from both the left side and the right side and avoid overfitting. Again, with the amount of data we have, we decide to go with the simpler form of model that captures the processes that we are interested in. However, in the future we are interested in trying out these more sophisticated models that better capture more complex latent processes.

In the revised manuscript, we have included the two more sophisticated HMM models as described above, along with the results of model comparisons. We also added more justification to the HMM we used and we apologize for the failure on our part to clearly explain our methodology and analytic approaches, so we have revised the manuscript substantially to add clarity.

New text from the manuscript (page 33, line 808-821):

“To account for the effect of biased exploitation on the probability of transitioning between states, we also considered an unrestricted HMM model with no parameter tying (4 parameter ntHMM), where we treat exploiting the left side and exploiting the right side as two separate exploit states and allow differential transition probability to each exploit state. However, the ntHMM did not improve model fit with almost identical log-likelihood (2-parameter original HMM: log-likelihood = -1424.0; 4-parameter ntHMM: log-likelihood = -1423.8). Then we compared two models by calculating the AIC/BIC values for both models, which penalized extra parameters in the no parameter-tying HMM. The AIC test favored the original 2-parameter model (AIC: 2-parameter original HMM: AIC = 2976.1; 4-parameter ntHMM: AIC = 3103.5; relative likelihood (AIC weight) of the 4-parameter ntHMM < 10^-28). The BIC test also favored the simpler 2-parameter model (BIC: 2-parameter original HMM: BIC = 3562.8; 4-parameter ntHMM: BIC = 4276.9; relative likelihood (BIC weight) of the 4-parameter ioHMM < 10^-155). In the light of the model comparison results, we decided to fit the simpler 2-parameter HMM model.”

(1.3) Moreover, the model assumes that the mice had to pass through exploration in order to start exploiting a new option. Do authors have any evidence supporting this assumption? What will happen to the results if this assumption is lifted? Please comment.

We apologize for not providing clearer justification for this assumption in the manuscript. We have revised the text to better explain this assumption from a normative point of view and a computational point of view.

The utility of exploration is to maximize information about the environment, whereas the goal of exploitation is to maximize reward or gains (Mehlhorn et al., 2015). In either of these commonly used definitions of exploration, if an animal switches from a bout of exploiting one option to another option, that very first trial after switching should be exploratory because the outcome or reward contingency of that new option is unknown and that behavior of switching aims to gain information. Therefore, if this assumption is lifted, the states that were inferred from HMM may not align with the normative definition of exploration and exploitation.

A second reason is that allowing direct transition between exploit states will increase the number of parameters to estimate in the HMM model. As we mentioned above, we do not have enough data to accurately estimate the parameters that account for the probability of direct transition between exploit states (due to the low occurrence of such circumstances).

New text from the manuscript (page32, line 771-781):

“The model also assumed that the mice had to pass through exploration in order to start exploiting a new option, even if only for a single trial. This is because the utility of exploration is to maximize information about the environment, as defined in both animal foraging literature and reinforcement learning models (Mehlhorn et al., 2015). If an animal switches from a bout of exploiting one option to another option, that very first trial after switching should be exploratory because the outcome or reward contingency of that new option is unknown and that behavior of switching aims to gain information. Through fixing the emissions model, constraining the structure of the transmission matrix, and tying the parameters, the final HMM had only two free parameters: one corresponding to the probability of exploring, given exploration on the last trial, and one corresponding to the probability of exploiting, given exploitation on the last trial.”

2) Page 10, line 252: Please provide a more quantitative comparison of models’ choice behavior (and not just RLCK) and the animals’ behavior for all sessions. Also, there are no tick-marks on the y-axis.

Thank you for your feedback. To quantitative how well each RL model was predicting animals’ choices, we examined model agreement for each model, which is calculated as the average probability of choice being correctly predicted by the model p (choice|model). Then, we conducted a multiple comparison across model agreement of RL models (Supplemental Table 2, Figure 3C). The results suggested that the RLCK model was the most simple model that best predicted animals’ choices. The tick marks have been added to the figure.

New text from the manuscript (page 13, line 341-356):

“To quantify how well each RL model was at predicting animals’ choices, we measured the model agreement for each model, which was calculated as the probability of choice correctly predicted by the optimized model parameters for each model (Figure 3C). Then we conducted a multiple comparison across model agreement of RL models (test statistics reported in Supplemental Table 2). The results suggested that the RL models with parameter(s) that account for choice bias (RLCK, RLCKγ, RLCKη) were significantly better at predicting animals’ actual choices than the models that do not account for choice bias and non-RL models (random, noisy WSLS, RL, RLε). There was no significant difference in model agreement between RLCK, RLCKγ, and RLCKη. Based on the result of model comparison (AIC) and model agreement, we decided that the four-parameter RLCK model is the simplest, best-fit model that best predicted animal’ actual choices. Finally, to visualize how well the RLCK model was at predicting choices of animals with different learning performance, we plotted the simulated choices and actual choices against the matching law (Poling et al., 2011), which dictates that the probability of choice is proportional to the probability of reward. The figure showed that this four-parameter model was able to characterize animals’ choice behaviors regardless of the value-based learning performance (Supplemental Figure 3).”

3) How much overlap exists between the extracted latent dynamics from HMM and that of previously proposed models mentioned in the methods (Daw et al., 2006; Jepma and Nieuwenhuis, 2011; Pearson et al., 2009)? It would be helpful to show the extent that results from these different methods deviate/overlap with each other.

That is a very interesting question, we have included a side-by-side comparison between RL-labeled explore-exploit states and HMM-labeled states. Previous studies have used reinforcement learning related models to estimate the values underlying decision-making (much as we do here with the various versions of the RL models), then identify exploration as the choices that do not match the values predicted by the model. Although differences between the tasks (i.e. probabilistic rewards, rather than reward magnitudes drawn from gaussian distributions) mean we cannot directly apply the exact model Daw et al., 2006 developed for their task, we can examine the relationship between choices that do not maximize reward per our best-fitting RL model and the exploratory choices inferred from the HMM.

We first examined the correlation between explore-exploit states inferred by the HMM model and the RLCK model and found that states inferred by these two methods are moderately correlated (Supplemental Figure 2B, r_tet_ = 0.42). Both HMM-inferred states and RL inferred states showed similar effects on response time – response time was significantly longer during exploration than exploitation (RL label: paired t-test, t(31) = 2.08, p = 0.046; HMM label: t(31) = 3.66, p = 0.0009). However, the effect size of HMM labels on response time was over twice as big as that of RL labels (HMM: R^2^ = 0.30; RL: R^2^ = 0.12). This analysis highlights the overlaps of two methods (HMM and RL model) in inferring explore-exploit states but the HMM explained more variance in response time. We also calculated the standardized regression coefficients to measure how much of the response time is explained by states odellin by HMM model and RLCK model (Supplemental Figure 2C). Again, the result suggested that the HMM-inferred states explained significantly more variance in response time than the RL-inferred states in explaining response time.

While both models have demonstrated capacity to differentiate between states (HMM did a better job than the RL model), the RL models label choices that are inconsistent with model prediction as exploratory. However, exploration as a non-reward-maximizing goal should be orthogonal to reward instead of errors. A few recent papers have explained this problem in some depth (Ebitz et al., 2018; Wilson et al., 2021). We have included a Supplemental Figure panel and more text to compare these two state-labelling methods and we believe this change better justified our decision to use the HMM model to infer states and has strengthened our claims.

New text from the manuscript (page 6, line 147-222):

“To test this hypothesis, we first need a method to label each choice as an exploratory choice or an exploitative choice. […] Because this approach to infer exploration is agnostic to the generative computations and depends only on the temporal statistics of choices (Ebitz et al., 2018; Wilson et al., 2021; Ebitz et al., 2019; Ebitz et al., 2020), it is particularly ideal for circumstances like this one, where we suspect that the generative computations may differ across groups.”

4) Do authors see any differences between amount of exploration/exploitation at the beginning vs at the end of a session? How about across days? The fact that meta learning is observed, suggests that even in a single session, changes in the strategy of animals might be expected.

We agree with the reviewer that finding differences in exploration over time, either within or between sessions, would add an interesting wrinkle to the story, but unfortunately we did not observe any adaptation in the probability of exploration.

We have included the analyses in the text of the manuscript as follows (page 10, line 254-257):

“We calculated the probability of exploration early, mid, and late session. However, we failed to see changes in the amount of exploration within sessions. We also examined the amount of exploration across sessions, and we found that there were no changes in the amount of exploration across sessions in both males and females.”

It is difficult to know whether this null result means that there is no adaptation. Alternatively, we may just not be able to measure adaptation due to variability in reward schedules over time or across trials. However, we believe this is an important question for future research and appreciate the comment.